# Using metacommunity ecology to understand environmental metabolomes

Robert E. Danczak[1], Rosalie K. Chu [2], Sarah J. Fansler [1], Amy E. Goldman[1], Emily B. Graham [1], Malak M. Tfaily[2,3], Jason Toyoda [2] & James C. Stegen [1✉]

Environmental metabolomes are fundamentally coupled to microbially-linked biogeochemical processes within ecosystems. However, significant gaps exist in our understanding of their spatiotemporal organization, limiting our ability to uncover transferrable principles and predict ecosystem function. We propose that a theoretical paradigm, which integrates concepts from metacommunity ecology, is necessary to reveal underlying mechanisms governing metabolomes. We call this synthesis between ecology and metabolomics 'meta-metabolome ecology' and demonstrate its utility using a mass spectrometry dataset. We developed three relational metabolite dendrograms using molecular properties and putative biochemical transformations and performed ecological null modeling. Based upon null modeling results, we show that stochastic processes drove molecular properties while biochemical transformations were structured deterministically. We further suggest that potentially biochemically active metabolites were more deterministically assembled than less active metabolites. Understanding variation in the influences of stochasticity and determinism provides a way to focus attention on which meta-metabolomes and which parts of meta-metabolomes are most likely to be important to consider in mechanistic models. We propose that this paradigm will allow researchers to study the connections between ecological systems and their molecular processes in previously inaccessible detail.

[1] Pacific Northwest National Laboratory, Richland, WA, USA. [2] Environmental Molecular Sciences Laboratory, Richland, WA, USA. [3] Department of Soil, Water and Environmental Science, University of Arizona, Tucson, AZ, USA. ✉email: James.Stegen@pnnl.gov

Environmental metabolomics enables the investigation of the metabolic processes and interactions occurring within an ecosystem and can provide deep insight into ongoing biogeochemical cycles[1–4]. High-resolution mass spectrometric techniques, like Orbitrap-MS, Fourier transform ion cyclotron resonance MS (FTICR-MS), and ion mobility spectrometry MS (IMS-MS), among others have allowed researchers to investigate the individual carbon compounds that constitute natural organic matter (NOM)[1,5–7]. As these studies increasingly become spatiotemporally resolved, an investigation of the underlying processes driving metabolome variability becomes necessary in order to develop transitive principles and enhance predictive capabilities across ecosystems. While many metabolomics studies have used multivariate methods to identify if differences exist between metabolomes[7–12], they have limited capacity to reveal processes that constrain or promote variation[13–15]. Initial applications of ecological theory to NOM have revealed interlinkages between microbial community dynamics and molecular formula functional diversity[16–18]. To better understand the processes governing metabolome composition, however, we propose a direct integration of concepts and tools developed in metacommunity ecology, the study of communities across scales[19], with environmental metabolomics. This will allow us to investigate mechanisms underlying spatiotemporal dynamics of metabolites as a conceptual analog to ecological metacommunities, and uncover transferable principles related to NOM organization. Our specific goals are to (1) explore the conceptual parallels between environmental metabolite assemblages and ecological metacommunities and (2) demonstrate how concepts and analysis tools from metacommunity ecology can be used to study environmental metabolites using an example FTICR-MS dataset from the Columbia River and adjacent riverbed. We demonstrate that novel insights are revealed by applying metacommunity ecology theory to environmental metabolomics and contend that this constitutes the line of inquiry called 'meta-metabolome ecology.' As a parallel to metacommunity ecology, meta-metabolome ecology is focused on studying multiple metabolite assemblages connected through spatial processes (e.g., advective transport).

Ecological communities are assembled through the collective action of random birth and death events, dispersal of individuals, deterministic factors, which affect the relative fitness of taxa, and the evolutionary processes of diversification and extinction[20]. Therefore, any community can be assumed to be the outcome of many different ecological and evolutionary assembly processes experienced throughout time and space[21]. Similarly, ecosystem metabolomes (i.e., assemblages of 'metabolites' which are discrete organic compounds that can be subject to biological activity) represent the collective outcome of historical processes that have resulted in the gain, loss, and transformation of individual metabolites. In our conceptualization, ecosystem metabolomes are influenced by and comprised of molecules generated within or transported into a given system. We refer to ecosystem metabolomes connected via spatial processes as 'meta-metabolomes,' akin to metacommunities. We suggest the associated set of historical processes shares many parallels with those governing ecological community assembly.

As with ecological communities, we hypothesize that metabolite assemblages are subject to mixtures of deterministic and stochastic assembly processes. Deterministic factors which influence ecological community composition occur due to systematic differences in birth and death rates among resident taxa (i.e., some taxa more successfully produce offspring than others given some environmental constraint)[13,22–24]. For example, differential abilities across taxa to scavenge nutrients can lead to a deterministic change in community structure through time[25]. Similarly,

individual metabolites within metabolite assemblages will undergo fluctuations in production rate (analogous to birth rate) and degradation rate (analogous to death rate), driven by abiotic or biotic transformations[1,4,11,26]. Distinct from biological systems, however, metabolite 'death' is inherently coupled to metabolite 'birth' as one metabolite is transformed into another. How this coupling influences our conceptual understanding deserves elucidation beyond what we can examine here. Nonetheless, any factors that adjust production/degradation rates could lead to a deterministic shift in the composition of the metabolite assemblage. For example, the preference of microorganisms for organic nitrogen in a nitrogen-limited environment could deterministically shift metabolite composition such that biogeochemical hotspots become characterized by nitrogenous metabolites[1]. The persistence of low oxidation state organic carbon within anaerobic environments due to the preferential consumption of more thermodynamically available compounds represents another potentially deterministic shift[27]. Increasingly homogenous DOM functional diversity derived from marine environments linked to ongoing degradation could also represent another deterministic impact[17].

Aside from purely deterministic environmental pressures, dispersal processes can strongly influence ecological community composition through variations in organismal exchange rates[20]. When the rate of exchange between two separated communities is high, these communities can become more homogeneous with respect to each other due to 'mass effects' (e.g., the flow of individuals from high to low population size)[28–30]. Although metabolites are independently governed by passive movement, high rates of metabolite exchange (e.g., via advective hydrologic transport or high rates of vector movement) could conceivably homogenize assemblages through space. The potential for vectors (e.g., microbes) to move and influence metabolite dispersal points to an intriguing interaction between biology and chemistry that deserves attention beyond the scope of this manuscript, but can likely be investigated using our framework. At the other extreme, dispersal rates can be low enough that ecological communities diverge in composition as a result of stochastic ecological drift (i.e., random fluctuations in birth and death rates). If there is a lack of temporally or spatially consistent factors driving variation in metabolite production/degradation rates, a dynamic similar to ecological drift should emerge, referred to here as 'metabolite drift' where unstructured compositional deviations occur in metabolite assembles. For example, if dispersal is limiting and any thermodynamic or nutritional requirement is too weak, we could expect a 'metabolite drift' signal. Therefore, low exchange rates of metabolites can lead to outcomes that are conceptually analogous to dispersal limitation in ecological communities.

Despite the many parallels between ecological communities and metabolite profiles, there are important differences. For example, no clear parallels can be drawn between the evolutionary processes influencing ecological communities and metabolite assemblages. Specifically, the rates of diversification and extinction which affect the relationship between the regional and local species pools do not translate to metabolites[20]. This is because metabolites cannot reproduce or pass along heritable traits. For example, while degradation products may have similar biochemical traits as evidenced by studies measuring DOM functional diversity[16–18], this is not always explicitly true (i.e., dephosphorylation, sulfurylation) and doesn't encapsulate all synthesis or degradation products. Instead, evolutionary processes could exert an indirect effect on the metabolome by acting upon biological taxa participating in production or degradation of individual metabolites. While we focus on conceptual parallels to deterministic and stochastic ecological factors, we can envision an extension of our framework, which integrates indirect

evolutionary forces into understanding spatiotemporal dynamics of ecosystem metabolomes.

Another discrepancy between ecological communities and metabolite assemblages exists with species interactions. In some situations, there is good correspondence. For example, in both ecological communities and metabolite assemblages, members (i.e., biological species or metabolites) will exert indirect and direct pressures on other members, through interactions like predation of individuals or the complexation of metabolites, respectively[31–33]. However, unlike ecological communities, metabolite assemblages are incapable of competitive exclusion such that mechanisms of coexistence do not have direct analogs within metabolite assemblages[20]. Despite these departures from traditional ecological communities, we believe that our hypothesis that metabolite assemblages experience stochastic and deterministic processes is conceptually supported. We assert that examining metabolite profiles using metacommunity ecological concepts and tools provides a perspective on ecosystem metabolites that will enable further conceptual and mechanistic understanding. This knowledge is a critical element of ongoing efforts to improve process-based predictive models of ecosystem function (e.g., reactive transport models of river corridors or soil carbon models) that underlie broader Earth system function.

Given the similarities between the deterministic and stochastic processes acting upon ecological communities and metabolite assemblages, we propose that ecological concepts and tools can be applied to metabolomes to gain insight into factors governing meta-metabolome spatiotemporal dynamics. We specifically attempt to understand the assembly processes governing the composition of metabolite assemblages because they are directly analogous to 'community assembly processes' governing the composition of ecological metacommunities. We focus on applying phylogeny-informed diversity metrics and phylogenetic-based null models that complement traditional multivariate statistics by revealing assembly processes[13,24,34]. The null modeling approach adapted here for metabolite assemblages has been shown to provide robust estimates for the relative influences of different assembly processes[13,24,35–38]. Due to the absence of explicit metabolite phylogeny, we use trait-based dendrograms that represent shared and divergent metabolite properties[39]. As in ecological functional trait or phylogenetic α-diversity and β-diversity analyses, the dendrogram approach provides information beyond simple taxonomic (or metabolite) assignments[40–42]. We demonstrate how relational dendrograms can be used to study metabolome α-diversity and β-diversity, and how they can be used with null model analyses to reveal assembly processes governing the composition of meta-metabolomes.

## Results

**An example set of metabolite assemblages and microbial communities.** We use metabolite data from the Columbia River corridor to provide an example of how to use a dendrogram-based framework to study the processes influencing metabolite assemblages. In brief, samples of river water and pore water were collected on November 19, 2017 from five locations (Supplementary Fig. 1, Supplementary Table 1) along the mainstem Columbia River in Washington State across a ~1 km transect running along the shoreline. This part of the Columbia River is in an arid region, is dam regulated, is predominantly gravel bedded, experiences significant groundwater-surface water mixing in pore fluids, and has been studied and described extensively[36,43,44]. At each location, filtered river water and subsurface pore water were collected; one replicate of river water was collected, and three pore water samples were collected from 30 cm depth within a 1 m2 area using 0.25-inch diameter

sampling tubes. Samples were analyzed using FTICR-MS at the Environmental Molecular Sciences Laboratory using previously established methods. The raw FTICR-MS data were processed according to established methods to (1) identify peaks from the mass spectra that correspond to unique metabolites identified by their unique mass, (2) calibrate peak/metabolite masses against a standard set of known metabolites, and (3) assign molecular formula based on the Compound Identification Algorithm (CIA)[45,46]. Further data analyses are described below in the subsections that use the associated analysis. In addition, water samples were analyzed for basic geochemical parameters (i.e., dissolved organic carbon concentration, specific conductivity, and major anions and cations). We extracted DNA from the filters used to collect aqueous samples and characterized associated microbial communities using 16 S rRNA gene sequencing and associated data processing to pick operational taxonomic units and generate a phylogenetic tree.

**Building metabolite dendrograms.** Tools and metrics in metacommunity ecology often leverage relational information such as among-species evolutionary relatedness or functional trait similarities, allowing researchers to reveal the balance among stochastic and deterministic assembly processes[23,35,41,42,47,48]. While metabolites do not have genetic sequence information, their characteristics can be approached in a way that is analogous to the functional trait approach in ecological analyses[39,49]. Unlike multivariate dendrograms typically used within metabolomics studies (e.g., Tfaily et al. 2018)[7], these dendrograms represent relationships between metabolites and not samples. To this end, we developed and evaluated three methods of measuring trait-like relational information between different chemical compounds using two different information sets: molecular characteristics and biochemical transformations (Fig. 1, Supplementary Fig. 2, Supplementary Data 1–3).

First, we generated a molecular characteristics dendrogram (MCD) which integrates elemental composition (e.g., C-, H-, O-, N-, S-, P-content) and derived statistics (i.e., aromaticity index, double-bond equivalents, etc.) similar to principles outlined in compound classification studies[50–56] or in NOM functional diversity analyses[16–18,57]. Next, we created a transformation-based dendrogram (TD) using putative biochemical transformations identified by aligning mass differences to a database of known transformations[1–3,9,51,58,59] (Supplementary Data 4). Finally, we made the transformation-weighted characteristics dendrogram (TWCD), which is a combination of the MCD and TD (Supplementary Fig. 2). Given each dendrogram method incorporates FTICR-MS peaks differently, the number of peaks incorporated into downstream analyses also varies (Fig. 2a, Supplementary Fig. 3; see Supplement for details). For example, while the MCD incorporates all assigned molecular formula (~14% of observed peaks in this dataset), the TD can gain access to a broader range of peaks because formulas are not required (~72.5% of observed peaks) (Supplementary Fig. 3). While there is a discrepancy between these approaches, this is due to inefficient formula assignment of FTICR-MS data and will vary from dataset to dataset, and with improved formula assignment tools[60]. Detailed differences between these dendrograms are explored in the Supplement, but each resulted in different metabolite clustering patterns that help provide deeper insight into ecosystem assembly. We suggest that while other approaches to estimating dendrograms from metabolite data exist, the MCD, TD, and TWCD provide a complementary set of methods that are useful for studying the spatiotemporal organization of meta-metabolomes.

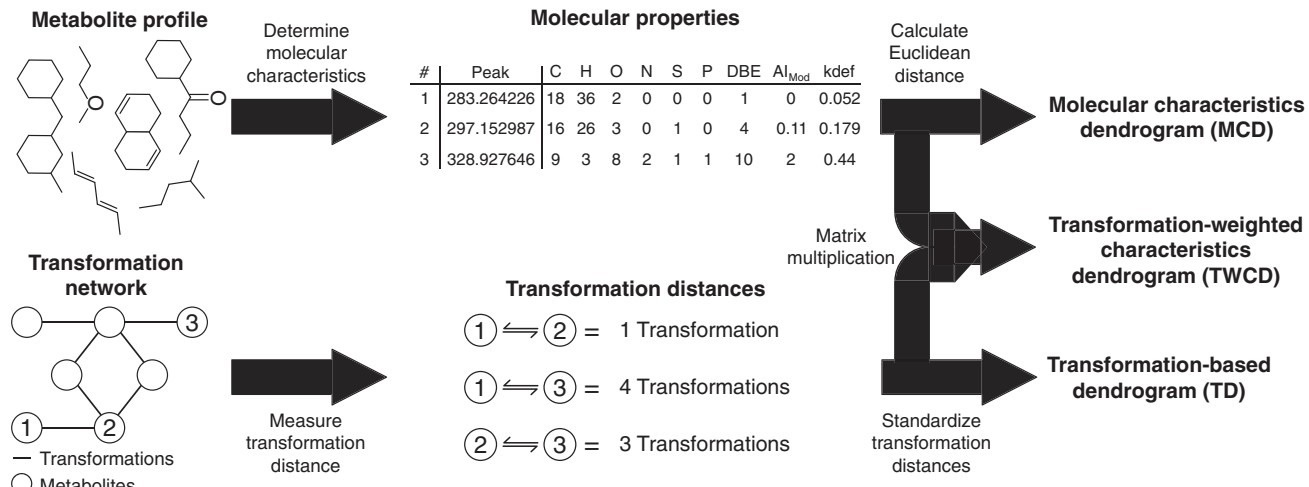

**Fig. 1 Figure summarizing the steps necessary to create the three dendrograms used throughout this manuscript.** The top path (Molecular Characteristics Dendrogram or MCD) demonstrates the relational information provided by molecular properties, like elemental composition and aromaticity index, while the bottom path (Transformation-based Dendrogram or TD) emphasizes the relationships driven by potential biochemical transformation networks. The middle path (Transformation-Weighted Characteristics Dendrogram or TWCD) is a combination of information provided by the top and both paths. All metabolites in the transformation network would have been identified; the numbered metabolites are used to demonstrate the approach. Definition of acronyms under molecular properties: C, H, O, N, S, and P are elemental counts; DBE is double-bond equivalents; $AI_{Mod}$ is modified aromaticity index; and kdef is Kendrick defect.

Importantly, data collected using an FTICR-MS will include information about any ionizable compound, not just those associated with biological systems[61]. Despite this potential limitation, previous studies have demonstrated that this type of data still contains biogeochemically relevant information[1,2,4,16,17]. Therefore, the three dendrograms described above can resolve the potential relationships between molecular formula based upon a point of view, which is agnostic to a molecular formula's source (MCD), a point of view which encompasses a putative biochemical point of view (TD), and an integrated view (TWCD). As with many of the tools described in this manuscript, the lack of explicit biological information provides two key benefits. First, it embraces the perspective that there is inherent value in investigating the processes, which give rise to all molecular formula, not just those involved in microbiologically mediated reactions. This allows for evaluation of intrinsic metabolite assemblage turnover without requiring potentially inaccurate biological assumptions. Second, it allows for the coupling of meta-metabolome ecology with other multi-omics data types. This approach minimizes errors that could occur by assigning the sources for molecular formula and associated transformations a priori, and allows understanding to be derived *a posteriori* through coupling to additional data types.

**A quick note about phylogenetic signals.** In order to ensure that a phylogenetic tree accurately captures the functional trait information of an ecological system, a test for a phylogenetic signal must be first performed[13,24,62–64]. Once a phylogenetic signal is confirmed, a range of ecological null models can be used to infer community assembly processes[13]. Within many ecosystems, this can be measured by calculating one of many phylogenetic signal metrics using average trait values[63]; in microbial systems where said trait values are not as readily available, estimated niche values are calculated based upon abundance and environmental data instead[13,64]. However, when functional trait dendrograms are used instead of a phylogenetic tree, a phylogenetic signal is unnecessary as the trait relationships are already built into the framework[39]. Given that the three proposed dendrograms are closely aligned to functional trait dendrograms (i.e.,

molecular formula properties and putative biochemical relationships)[16,17], phylogenetic signal is unnecessary when implementing associated null models.

**Using metabolite dendrograms to study metabolite diversity and assembly processes.** From a practical perspective, the three dendrograms provide a foundation for studying metabolite assemblages with ecological tools that traditionally use phylogenetic or functional trait data. For example, below we show how metabolomes can be studied using metrics associated with richness (Faith's PD, UniFrac), overall divergence (MPD), and nearest neighbor divergence (MNTD)[42,47,48,65]. As a parallel to ecological analyses, these metrics can be used to study the spatial and temporal organization of meta-metabolomes.

Many ecological studies track trait dynamics or utilize identity-based (i.e., taxonomic) analyses such as Bray–Curtis dissimilarity to infer ongoing ecosystem processes[66,67]. There are, however, exciting opportunities to go further by using additional tools from metacommunity ecology that are designed to infer and quantify assembly processes. Null models represent one set of tools that provide additional insight and complement traditional α-diversity and β-diversity analyses. By applying commonly used phylogenetic null models, we can investigate the processes responsible for structuring metabolite assemblages. First, to assess whether α-diversity was more or less structured than would be expected by random chance, we calculated both the net relatedness index (NRI) and nearest taxon index (NTI), which are z-scores quantifying deviation from null models for MPD and MNTD respectively[23,65]. For both these metrics, positive values indicate clustering within the dendrogram while negative values signify overdispersion[65].

Ranging from cold weather adaptation in forests[68], labile carbon degradation in bacterial communities[69], or host range/soil adaptations in root-associated mycobiomes[70], these metrics have revealed patterns in phylogenetic trait conservation through different phylogenetic lineages[71]. Despite examining different ecosystems and scales, a common framework enabled researchers to develop consistent conceptual conclusions. In turn, these null models should provide a similar framework for metabolite

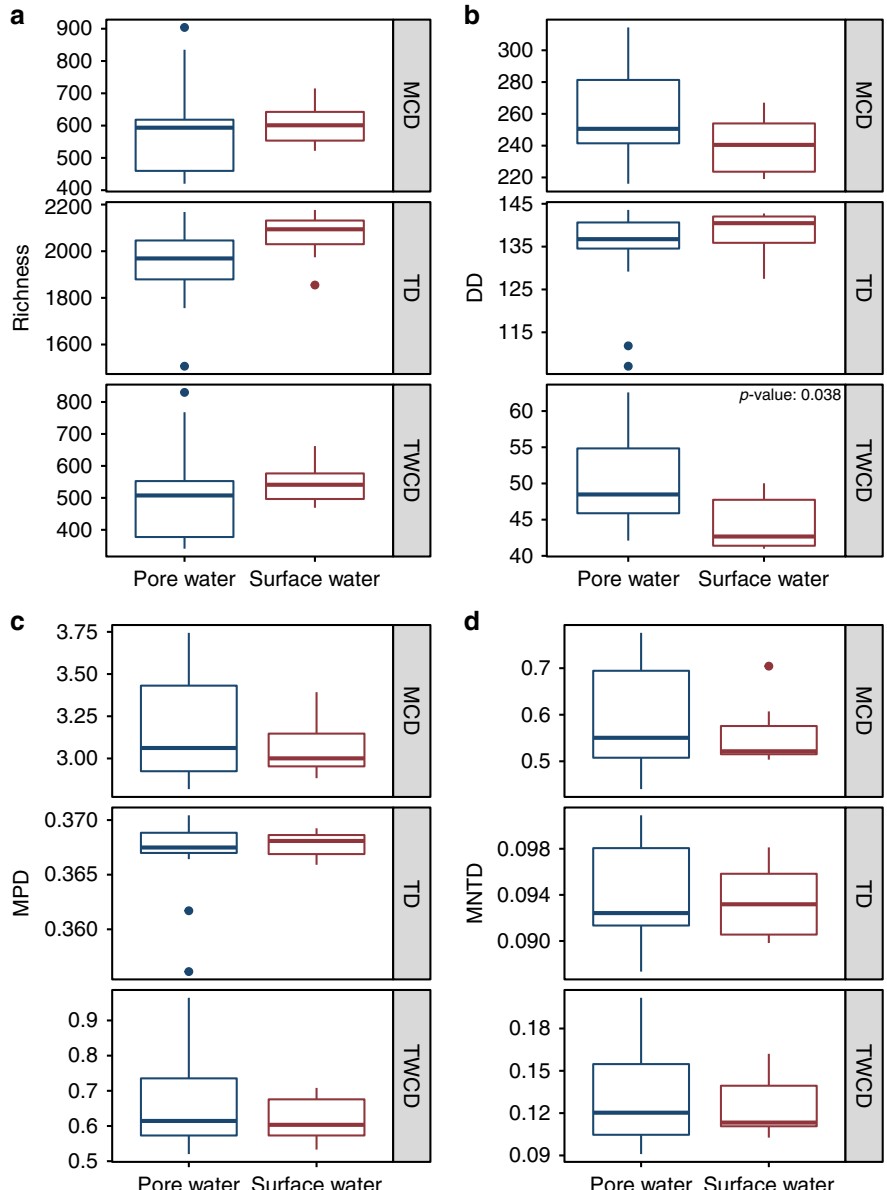

**Fig. 2 Alpha diversity boxplots for the metabolite data. a** Richness (akin to metabolite count). **b** Dendrogram Diversity (DD) which is analogous to Faith's Phylogenetic Diversity (PD). **c** Mean Pairwise Distance (MPD). **d** Mean Nearest Taxon Distance (MNTD). Two-sided Mann–Whitney U tests (Surface water $n = 7$, Pore water $n = 14$) determined that only the TWCD-DD comparison was significant; the $p$ value is indicated within the figure. Each panel represents metrics calculated for the corresponding metabolite dendrogram (e.g., MCD, TD, and TWCD). Boxes represent the 1st and 3rd quartiles, the horizontal line within the box represents the median, the vertical lines represent extreme values calculated based on the interquartile range, and the points are potential outliers.

assemblages, with varied interpretations dependent upon the dendrogram. For example, overdispersion observed on the MCD might suggest broadly distributed thermodynamic properties while it could indicate biochemically disconnected peaks on the TD. Such analyses will allow researchers to ask and answer questions regarding the development of meta-metabolomes.

To further explore the ecological assembly processes structuring metabolite profiles, we calculated the β-nearest taxon index (βNTI; detailed extensively in Stegen et al. 2012, 2015). This metric compares the observed β-mean nearest taxon distance (βMNTD) between two communities to a null expectation generated by breaking observed dendrogram associations. While typically informed using abundance data, this null model still produces useful information with presence/absence data. When a comparison between two ecological communities significantly

deviates from the null expectation (indicated by $|\beta NTI| > 2$), we infer that some deterministic process is responsible for the observed pattern. These deterministic processes can be further separated into those which drive a divergence between communities, termed 'variable selection' (indicated by $\beta NTI > 2$), and those which drive a convergence between communities, termed 'homogeneous selection' (indicated by $\beta NTI < -2$). In a biological context, for example, these processes could result in a microbial community being driven toward a common configuration due to homogenous selection resulting from primary succession within soil, or being driven toward divergent compositions due to variable selection resulting from varied organic matter[62]. If the pairwise comparison instead mirrors the null expectation (indicated by $|\beta NTI| < 2$), we infer that stochastic processes drive observed differences. These stochastic processes can be

further distinguished using an identity-based ("taxonomic") null model, like Raup–Crick, which is able to disentangle dispersal limitation from homogenizing dispersal (i.e., mass effects) when used in conjunction with βNTI.

Previous studies that combined βNTI and Raup–Crick have revealed significant variation in the relative influences of different community assembly processes among systems. This previous work spans a broad range of systems such as the subsurface[9,24,35,37], soils[3,62,72,73], human microbiome[74], marine[75]. Other studies[76,77] have applied these methods in cross-ecosystem analyses to understand common pressures. While each of these studies posed distinct questions, they are united by an emphasis on understanding the relative contributions of different assembly processes and linking assembly processes to other system features (e.g., redox conditions, succession, abiotic dynamics, ecosystem function, human society, etc.). Having a common conceptual grounding across studies provides an opportunity to investigate assembly processes affecting metabolite assemblages, and to develop theory that applies across and within ecosystems and spatiotemporal scales. Moreover, this common theory can be used to study ecological communities (e.g., microbes) and the metabolites they transform using the same framework, as previously performed with bacteria and viruses[78]. The degree of coordination between assembly processes can be subsequently related to microbial processes, DOM components, and environmental factors to reveal variables important to convergent and divergent assembly, in turn providing insight into those factors underlying biogeochemistry. These significant ecosystem variables can then be used to inform the mechanistic models that represent organisms and metabolites within dynamic ecosystems (e.g., reactive transport models).

**Example use of the dendrogram-based framework**. We use data from the Columbia River corridor to provide an example of how to use the dendrogram-based framework to study the processes influencing metabolite assemblages. Samples were collected from different environments (i.e., river water and subsurface pore water) but in a system with significant hydrologic connectivity between these environments[79]. We use the FTICR-MS data discussed earlier to explore within-assemblage diversity (i.e., α-diversity) and between-assemblage compositional differences (i.e., β-diversity). These analyses are pursued with and without dendrogram-based quantification to compare insights between traditional approaches and dendrogram-enabled analyses. In addition, we primarily use dendrogram-based null modeling (e.g., βNTI) to investigate assembly processes, though we later combine it with dendrogram-free null modeling (e.g., Raup–Crick) to expand our conclusions. We expect that the distinct river water and pore water environments will lead to deterministic signatures associated with variable selection when comparing metabolomes across these environments. This expectation is due to differential NOM processing capabilities between surface and pore water, previously observed in our field system[1,2,36]. We further expect to observe some contribution from homogenizing dispersal due to significant hydrologic connectivity, and thus metabolite mixing, within and across surface and pore water in the field system[36,43]. In addition, we show how null model outcomes associated with metabolite assemblages can be related to null model outcomes associated with microbial communities. This provides an opportunity to evaluate the degree to which assembly processes are coordinated between microbial communities and the metabolites they produce and consume. This represents a conceptual unification across ecological communities, the resources they depend on, and the influences they have over environmental systems. It is important to recognize that the sample set used here is for demonstration purposes and is therefore relatively small to facilitate straightforward interpretation. We expect different analysis outcomes when the methods developed here are applied to other sample sets and other environmental systems.

**Metabolome α-diversity is deterministically organized**. Many metabolomic studies employ common multivariate statistics (e.g., ordinations) to determine whether differences exist between samples or sample groups[7,8,12]. While this can provide useful insights into similarities between samples, these methods do not incorporate among-metabolite relational information. Just as in ecological metacommunities, integrating relational information (e.g., phylogenetic or functional trait relationships) expands the breadth of inquiries one can pursue. The dendrogram-based approach developed here allows relationally informed α-diversity metrics to be applied to metabolite assemblages and can be used to investigate patterns driven by shared molecular characteristics and biochemical transformations.

Dendrogram Diversity (DD), directly analogous to Faith's Phylogenetic Diversity[47], is a relationally informed metric that quantifies the total dendrogram branch length occupied by a given metabolome. Higher values indicate metabolomes that span a broader range of molecular properties (MCD), that span more broadly across the biochemical transformation network (TD), or both (TWCD). The TWCD values for DD were significantly higher for pore water than river water, but the MCD- and TD-based DD did not differ between pore and river water (Fig. 2b). Two additional α-diversity metrics, mean pairwise distance (MPD) and mean nearest taxonomic distance (MNTD), revealed that pore water and river water metabolites share similar dendrogram topologies (i.e., compounds within a given group have similar branch lengths to other compounds) (Fig. 2c, d).

These dendrogram-based α-diversity metrics indicate that pore water metabolites are slightly more diverse in that they span a broader range of molecular properties. However, pore and river water metabolites are equally diverse with respect to potential biochemical transformation connections. This highlights that multiple dimensions of diversity exist within ecosystem metabolomes, each providing a different window into metabolome organization. The combination of dendrogram-free (e.g., number of unique metabolites) and dendrogram-based (e.g., DD) analyses provides an approach to investigate more dimensions of metabolome diversity than would be possible otherwise. Rather than focusing purely on between-group differences in molecular stoichiometry or other properties (e.g., thermodynamics), relationally informed α-diversity metrics allow questions related to the organization of metabolite assemblages to be assessed. For example, "How consistent are metabolite richness and DD?" or "How do DD values obtained from MCDs compare to those from TDs across systems?" can now be interrogated. Answering these questions will help reveal which abiotic or biotic features drive molecular formula and biochemical variability across different spatiotemporal scales and help highlight which DOM components should be included in mechanistic models. General patterns, which emerge from cross-system analyses, could point to transferable principles that could be integrated into mechanistic models linking metabolite chemistry to microbial and biogeochemical function.

To go beyond direct characterization of α-diversity, null modeling is often used in ecology to evaluate whether community structure deviates from a stochastic expectation[80]. A broad range of informative null model analyses can be used when a phylogenetic or relational dendrogram is available. In addition, α-diversity phylogenetic null models provide opportunities to evaluate questions not accessible via analyses that do not use

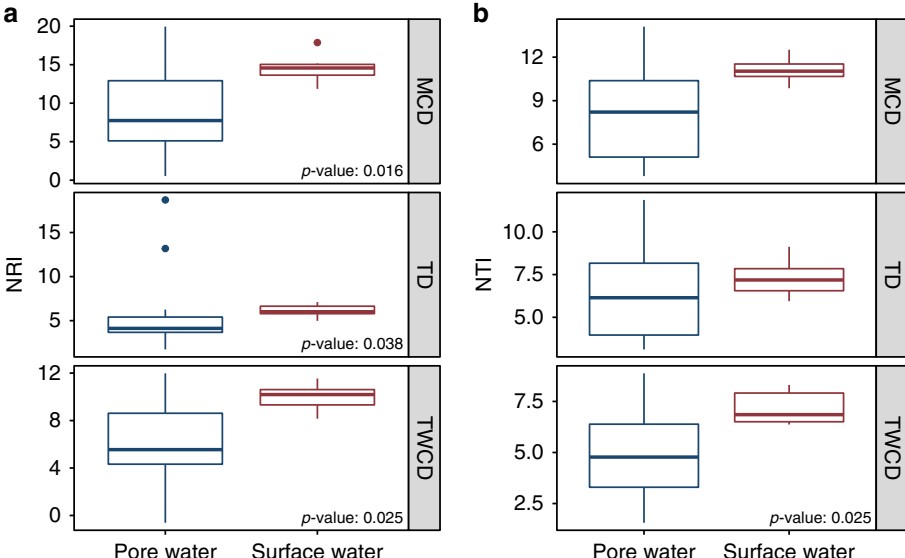

**Fig. 3 Alpha diversity null modeling results for the metabolite data. a** Net Relatedness Index (NRI). **b** Nearest Taxon Index (NTI). If differences between surface water and pore water samples was significant as determined by a two-sided Mann–Whitney U test (Surface water $n = 7$, Pore water $n = 14$), the $p$ value is indicated within the plot. Each panel represents metrics calculated for the corresponding metabolite dendrogram (e.g., MCD, TD, and TWCD). Boxes represent the 1st and 3rd quartiles, the horizontal line within the box represents the median, the vertical lines represent extreme values calculated based on the interquartile range, and the points are potential outliers.

relational information. Outcomes can be used in a variety of ways, such as revealing whether a given community is phylogenetically over- or under-dispersed[23,42,65]. Common null models like net relatedness index (NRI) or nearest taxon index (NTI) can be extended to metabolite assemblages using one or more metabolite dendrograms. Analogous to applying these methods to ecological communities, the degree of tip- (NTI) or deeper-level (NRI) clustering or overdispersion in metabolite assemblages can be quantified.

Both NRI and NTI revealed that the pore water and river water metabolite assemblages had significantly more clustering than would be expected by random chance as indicated by high positive values (Fig. 3). This was consistent across all three dendrograms and indicates an important influence of deterministic assembly processes that constrain the composition of metabolite assemblages. Furthermore, river water metabolites had greater clustering than pore water in every analysis aside from the MCD- and TD-based NTI (Fig. 3). Interpreted in concert with the α-diversity patterns discussed above, the null model results suggest that the decreased TWCD-based DD within river water metabolites (relative to pore water) is driven by an increased amount of clustering at both the tip-level (NTI) and across deeper relationships (NRI). Greater clustering in the TWCD indicates the presence of finer-level metabolite groups that are highly similar to each other in terms of their molecular properties and shared biochemical transformations.

Examining metabolite assemblages through the lens of community ecology provides opportunities to generate conceptual outcomes that would be challenging with traditional multivariate analyses. Here, α-diversity analyses revealed both pore and surface water were deterministically organized despite divergent mechanisms (Fig. 3). More specifically, pore water metabolites were moreover-dispersed than the river water metabolites according to every NRI and one NTI analysis. This demonstrates that a systematic driver causes pore water metabolites (1) to span a broader range of molecular properties and (2) be separated by a larger number of biochemical transformations. Such differences in the molecular properties and biochemical transformation networks topologies between pore and river water indicates that different localized processes influence metabolite assemblages across the river corridor ecosystem. This suggests a need to understand variation in the mechanisms that underlie metabolome assembly processes. To this end, additional insights can be gained by taking further advantage of the conceptual unification of metabolomics and metacommunity ecology to evaluate the processes influencing variation in metabolome composition (i.e., β-diversity).

**Stochastic molecular properties and deterministic biochemistry.** As a complement to α-diversity, β-diversity is commonly evaluated to capture multivariate differences among ecological communities. Previous studies have also used dendrogram-free β-diversity metrics (e.g., Jaccard) to study differences among metabolite assemblages[4,9,11]. As with α-diversity, these metrics can be extended by utilizing relational information provided by a phylogeny or dendrogram[23,24,42,48]. This additional relational information enables quantitative evaluation of relative influences of stochastic and deterministic assembly processes influencing spatiotemporal variation in the composition of ecological communities or metabolite assemblages.

While quantitative evaluations of stochasticity and determinism are common within community ecology, they have not been pursued within metabolomics. Such analyses open conceptual domains focused on the processes causing spatiotemporal variation in metabolomes. For example, while it is well known that stochastic processes work in concert with deterministic processes to influence spatiotemporal variation in ecological communities[81], it is unknown whether stochastic processes have any significant influence over spatiotemporal variation in metabolite assemblages. Given the strong influence of metabolite assemblages over biogeochemical function[1,2,9,16], stochastic influences are likely to alter biogeochemical function in potentially unpredictable ways[82]. Furthermore, given that ecosystem metabolites are both resources for and products of microbial metabolism, strong influences of stochasticity over metabolomes may cascade into microbial community assembly or indicate highly variable microbial metabolic processes at spatial

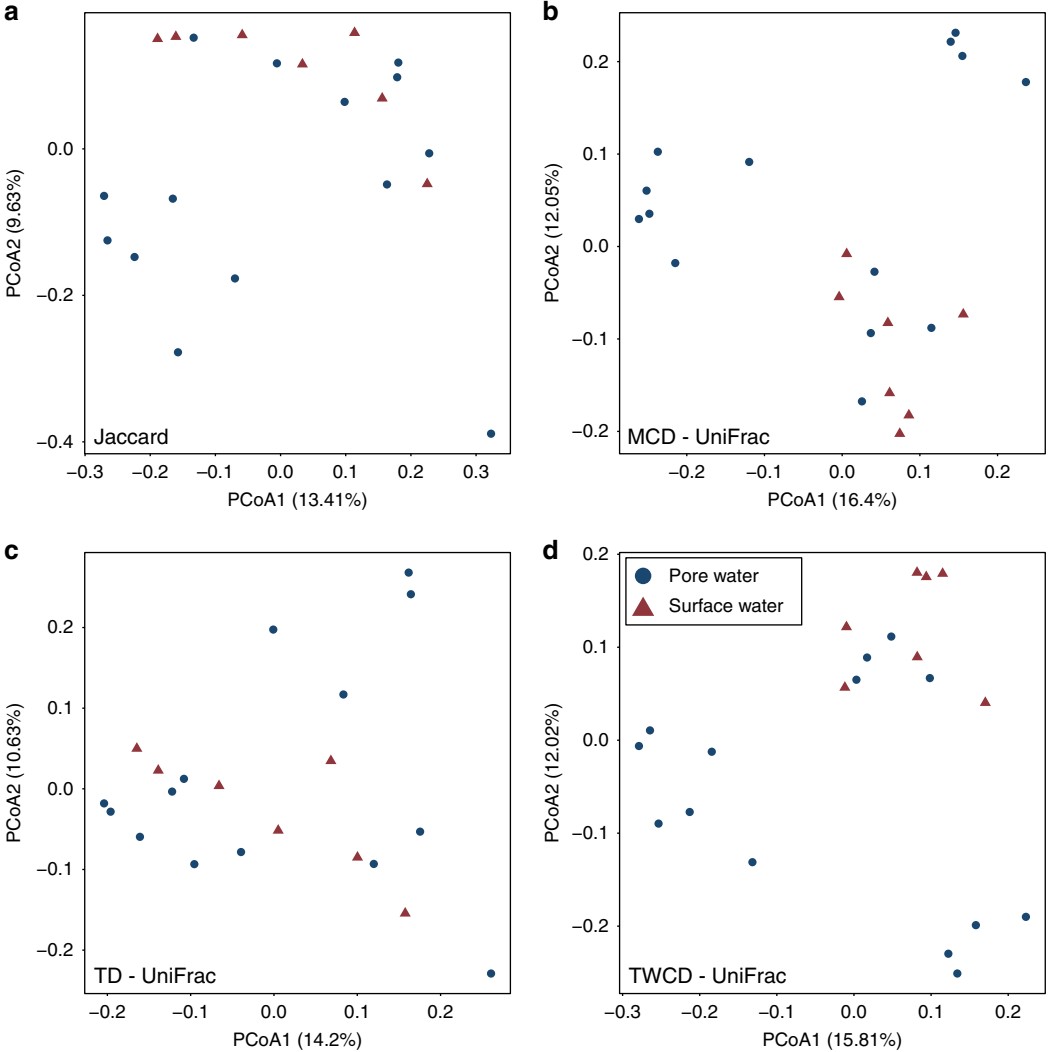

**Fig. 4 Beta-diversity ordination plots. a** Jaccard Dissimiarlity-based Principal Coordinate Analysis (PCoA). **b** UniFrac PCoA generated using the MCD. **c** UniFrac PCoA generated using the TD. **d** UniFrac PCoA generated using the TWCD.

scales below the sample volume. These examples only represent some types of scientific inquiry that can be opened up by examining spatial and/or temporal variation in metabolome composition (i.e., β-diversity) through the lens of meta-community ecology.

To study metabolome β-diversity, we examined both dendrogram-free and dendrogram-based metrics to provide the deepest conceptual insights. Using a dendrogram-free approach, β-diversity results from our dataset revealed greater differences than the α-diversity analyses. A Jaccard-based principal coordinate analysis (PCoA) revealed the existence of two clusters, with a PERMANOVA analysis revealing significant difference between surface and pore water (Pseudo-F: 1.48, $p$ value: 0.018; Fig. 4a). Incorporating relational information provided by the metabolite dendrograms resulted in the emergence of more defined clusters while maintaining significant differences. Here, relational information was integrated using unweighted UniFrac, which compares the number of shared and unshared branch lengths between two assemblages[48]. Three discrete clusters, which were not observed in Jaccard-based analysis emerged, when using either the MCD or TWCD, but not the TD (Fig. 4). This highlights the deeper level of information provided when considering the molecular and

biochemical relationships among metabolites, similar to the use of phylogenetic or functional trait information, and indicates that molecular properties are conserved within particular sets of metabolite assemblages. We infer that there are consistent biotic and/or abiotic processes acting to constrain molecular properties across subsets of metabolite assemblages, but not the biochemical transformations. UniFrac analyses, however, are not capable of identifying the relative contributions of stochastic and deterministic processes, which can instead by parsed out through the use of null models.

Similar to α-diversity, using relational information (e.g., phylogenies or dendrograms) with β-diversity null modeling can reveal the relative influences of stochastic and deterministic processes over spatiotemporal variation in the composition of ecological communities and metabolite assemblages. Given that phylogenetic analyses of microbial communities often use the beta nearest taxon index (βNTI) null model[2,3,38], we used it here to study metabolite assemblages. We encourage follow-on studies to explore patterns that emerge from different null modeling approaches applied to meta-metabolomes, as different metrics ask different questions, provide different pieces of information within the broader reality of a given system, and can therefore be used together to provide deeper inferences.

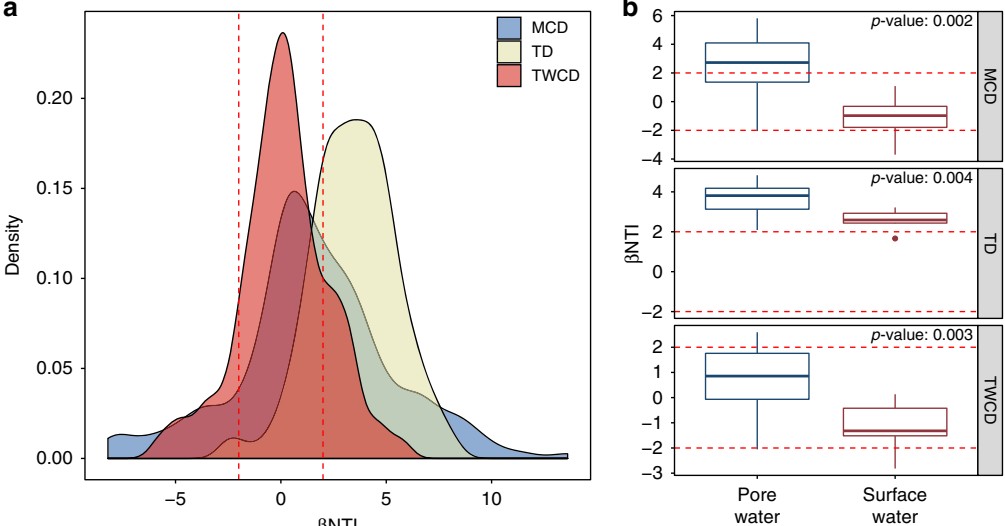

**Fig. 5 Beta-diversity null modeling results. a** Density plot of βNTI results for all comparisons. **b** Boxplots of average within-scale βNTI results (e.g., only pore water-to-pore water comparisons). Dashed red lines indicate the assembly process thresholds: βNTI < −2 represents homogenous selection, βNTI > 2 indicates variable selection, and |βNTI| < 2 indicates stochastic assembly. Two-sided Mann–Whitney U tests (Surface water $n = 7$, Pore water $n = 14$) were used to reveal significant differences between pore water and surface water distributions in panel **b**, with corresponding $p$ values listed in the figure. Boxes represent the 1st and 3rd quartiles, the horizontal line within the box represents the median, the vertical lines represent extreme values calculated based on the interquartile range, and the points are potential outliers.

As reviewed above, βNTI is a pairwise metric and is estimated by quantifying the difference between the observed beta mean nearest taxon distance (βMNTD) and βMNTD expected under stochastic assembly. If a given pair of metabolite assemblages are significantly more different from each other than would be expected under stochastic assembly (indicated by βNTI > 2), we infer that deterministic processes have caused divergence in metabolome composition. This situation is referred to as 'variable selection' in the ecological literature. In contrast, if a pair of metabolite assemblages are more similar to each other than expected (indicated by βNTI < −2), we infer that that deterministic processes have constrained the composition of those assemblages to be similar. This situation is referred to as 'homogenous selection' in the ecological literature. Lastly, if differences between a pair of metabolite assemblages do not deviate significantly from the stochastic expectation (|βNTI| < 2), we infer that stochastic processes (i.e., mixing, unstructured/ inconsistent gains/losses of metabolites, etc.) are primarily responsible for the observed differences in metabolite assemblages. We note that other β-diversity null models could be used to study metabolite assemblages in conjunction with βNTI to reveal additional insights.

Applying the βNTI null model to our dataset revealed that a mixture of homogenous selection, stochastic processes, and variable selection structured spatiotemporal variation in metabolite assemblages (Fig. 5a). The influences of these assembly processes differed sharply between molecular property and biochemical transformation-based relationships. βNTI associated with the MCD or TWCD demonstrated that all three structuring processes influenced metabolite assemblages, though stochastic processes and variable selection dominated (Fig. 5a). Comparatively, most TD-based βNTI values were >2, indicating that variation in the topologies of metabolite biochemical transformation networks were predominantly governed by variable selection. As such, the molecular properties in both the pore and river water samples were governed primarily by stochastic processes while the organization of biochemical transformations were deterministic (Fig. 5a).

**Changing scales leads to additional insights accessible only through null modeling based on relational information.** A powerful aspect of null modeling is that one can evaluate the relative influences of stochastic and deterministic influences at different scales[19,25,83]. For example, one can reduce the spatial scale of analysis to study processes causing variation in composition within a given environmental condition and compare that to assembly processes operating at larger scales (i.e., across environments). Here we take advantage of this scale dependence to study how inferred assembly processes change when constraining analyses within pore or surface water. This provides important insights as there can be processes that drive variation within a given part of an ecosystem that are not responsible for variation across ecosystem components[25]. To the best of our knowledge, ecological null modeling is the only robust approach to reveal scale dependence in assembly processes[24,35].

Our null modeling analyses within pore or surface water indicate that different assembly processes operate within each compartment of the river corridor ecosystem to influence metabolite properties, but not biochemical relationships (Fig. 5b). Within pore water, the molecular properties of metabolite assemblages were predominantly governed by variable selection and had higher average βNTI values than the surface water ($p$ value: 0.002–0.004). Previous work has shown that increases in βNTI are due to increasingly strong variable selection[35,62]. We therefore infer that across sampled pore water locations there was a greater divergence in localized deterministic processes, relative to surface water. For example, previous work in the study system showed that dynamic (and spatially variable) groundwater-surface water mixing primed microbial respiration[2]. This is one mechanism among many that could lead to deterministically organized spatiotemporal variation across pore water metabolite assemblages. In contrast, within surface water, the molecular properties of metabolite assemblages were predominantly influenced by stochastic processes. This indicates a potentially strong influence of spatial processes within the surface water, such as significant mixing of metabolite assemblages across the sampled locations.

While assembly processes influencing molecular properties were scale dependent, variable selection consistently governed variation in biochemical relationships suggesting each metabolite assemblage in pore or surface water had a distinct transformation network topology. We infer that even when the molecular properties of metabolite assemblages are influenced by stochastic processes (e.g., mixing), the associated biochemical transformations are distinct due to localized processing (e.g., enzymatic degradation) and generation (e.g., metabolite excretion) of organic molecules. By combining the results across scales, we revealed that significant variation within pore water metabolite assemblage assembly processes was not substantial enough to cause deterministic divergence in the molecular properties of surface and pore water metabolite assemblages when analyzed together.

Scale dependence in assembly processes points to a key challenge for representing the properties of metabolite assemblages within predictive models (e.g., integrated hydro-biogeochemical reactive transport models)[84,85]. Such models will need to grapple with when, where, and how to represent detailed mechanisms governing spatiotemporal variation in metabolite assemblages. For example, while the molecular properties of pore water metabolite assemblages diverged from each other due to variation in deterministic processes, it is not clear that variation in those processes is strong enough to warrant representation in predictive models. In contrast, the localized processes appear to strongly influence biochemical transformations at all scales, pointing to the need for representation of associated mechanisms.

As pointed to above, the inferences revealed here through at-scale and between-scale null modeling represent only a few of the conceptual insights that can be gleaned by studying metabolite assemblages through the lens of meta-community ecology. One can parse metabolite assemblages into different kinds of groups based upon elemental composition (e.g., looking at only N containing compounds), thermodynamic properties, or activity within a biochemical network and evaluate variation in assembly processes across these groups. As a demonstration of how insights can be gained by studying assembly processes across metabolite groups, we next examine stochastic and deterministic assembly processes within putatively more or less biochemically active metabolite groups.

**Metabolites across the activity continuum are essential to the deterministic organization of biochemical transformation networks**. Metabolites were parsed into putatively more or less biochemically active groups to evaluate whether these groups experience differential assembly processes. This was accomplished by separating those metabolites which were involved in no transformations (less active group) from those which were involved in a comparatively large number (>40) of putative biochemical transformations (more active group). The cutoff of 40 transformations was based on a clear decrease in number of transformations above 40 (Supplementary Fig. 4), although this is an arbitrary threshold. While these groupings are not definitive in terms of relative biochemical activity due to being based on an inexhaustive transformation list and a lack of biological measurements, they allow for demonstration and preliminary investigation into whether assembly processes vary significantly across metabolites that are more or less biochemically connected. One may expect that more biochemically connected metabolites are also more active and thus more deterministically organized due to the greater potential for influences of spatial processes (e.g., advective mixing) over less active metabolites.

Null modeling results from this analysis revealed that putatively more active metabolites experienced stronger deterministic influences. The βNTI results from the MCD and TCWD

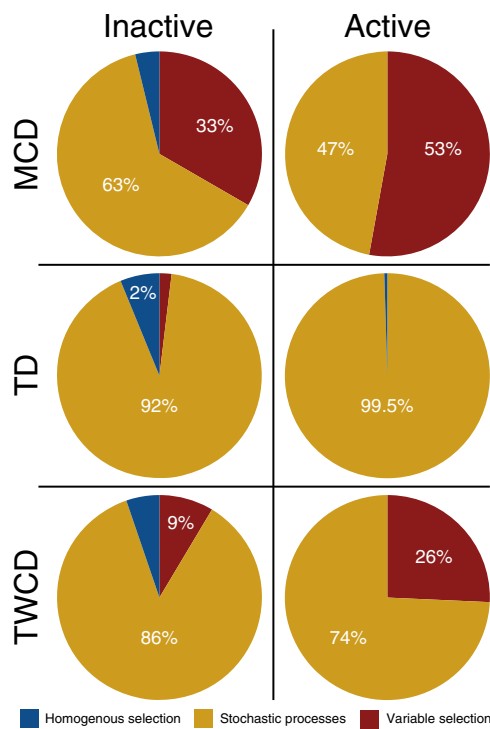

**Fig. 6 Ecological assembly processes divided by putative metabolite activity.** Pie charts illustrating the differences in ecological assembly processes for the putatively active (metabolites involved in >40 transformations) and inactive (metabolites involved in 0 transformations) fractions of the metabolite assemblages.

demonstrate that the more active metabolites were influenced by variable selection while the inactive metabolites were stochastically organized (Fig. 6). This suggests distinct underlying mechanisms governing the composition of more vs. less active metabolites. The TD-based null modeling, however, showed strong influences of stochastic processes over both more and less active metabolites. This is in direct contrast to the complete metabolite profiles where variable selection drove among-assemblage differences in biochemical relationships (Fig. 5). Given that the TD is based upon all possible transformations within a given dataset rather than only those observed in any one sample, the contrasting outcomes suggest that metabolites with relatively few transformation-based connections to other metabolites are key to localized, deterministic organization of biochemical transformations. As such, metabolites across the whole continuum of activity (and connectivity) levels are likely critical to the overall biogeochemical function of the system. Once combined with techniques that assess ecosystem function (i.e., enzyme measurements, respiration rates, omics techniques), this type of analysis will become even more informative. Beyond comparing assembly across scales, this framework provides an opportunity to study microbial communities and the ecosystem metabolites they interact with using the same conceptual foundation. For example, one can evaluate the degree to which there is coordination in the assembly processes influencing microbial communities and associated metabolite assemblages.

**Microbial and metabolomic assembly are not coordinated.** Microbial communities are a primary driver of ecosystem metabolite transformations and significantly impact rates of organic matter production and degradation[1,26,86,87]. In turn, there are likely to be dynamic feedbacks between metabolite assemblages

and microbial communities[1,9]. In order to approximate the extent of these interactions, we examined the relationship between the assembly processes acting upon microbial communities and metabolite assemblages. Given the potential for feedbacks, we expect that the relative influences of assembly processes governing microbial communities will correlate with the assembly processes influencing metabolite assemblages. For example, if the microbial community shifts due deterministic processes, we expect that associated metabolite assemblages will also shift deterministically if metabolite assemblages are governed by microbial composition and/or the same environmental factors influencing microbes. However, if microbes and metabolites are governed by different factors, deterministic assembly of one may not translate into deterministic assembly of the other.

Examining the microbial data in isolation reveals that river corridor communities were equally influenced by variable selection and stochastic processes (~43% of among-community variation in composition was due to each process). These relative influences mirror those estimated for the metabolite assemblages when using the MCD (~41% variable selection; ~46% stochastic). However, once the βNTI values associated with microbes and metabolomes were directly compared to each other using a Spearman-based Mantel correlation their apparent correspondence disappeared. Specifically, the Mantel statistic suggested there was no relationship (Mantel r: 0.223, $p$ value: 0.135).

While the lack of association of assembly processes between microbes and metabolites is inconsistent with our hypothesis, it points to complex factors influencing both metabolites and microbes. For example, while microbes are one part of the river corridor system that can influence metabolite composition, there are numerous other factors (e.g., vegetation, mineralogy, subsurface hydrology, photooxidation) that likely impact metabolite assemblages[26,86,88–90]. These nonmicrobial processes likely alter metabolite assemblages in a way that is not reflected in microbial community composition[88,90]. In addition, metabolite assemblages may change faster than microbial community composition, whereby there may be a closer association between metabolite assemblages and expressed metabolisms (e.g., metatranscriptomes) and/or relative changes in activity (e.g., relative rRNA content across taxa). Given that βNTI is inherently scale dependent due to its pairwise nature, analyses might require a change in scale to obtain the desired results. Lastly, these data do not indicate a lack of interaction between microbial communities and metabolite assemblages; instead, they suggest that microbial communities and metabolite assemblages may be driven by distinct sets of processes. In this context, individual molecular formula metrics or NOM functional diversity could provide complementary information regarding interactions[16]. By combining this framework with various omics techniques, we could begin to assess metabolic contributions to divergent assembly processes.

The degree to which there are associations between metabolite assemblages and various features of microbial communities is likely to vary through space and time. We posit that insights can be gained by studying the degree to which assembly processes are coordinated between metabolite assemblages and microbial communities. For example, a strong association between assembly processes influencing metabolites and community composition could indicate a relatively stable system with similar time scales of change for metabolites and microbial composition. Alternatively, strong associations between assembly processes influencing metabolites and expressed microbial metabolism, but not microbial composition, could indicate fast metabolite dynamics coordinated with rapid changes in microbial metabolism despite relatively slow changes in microbial composition. It will be fascinating to explore the associations among assembly processes

influencing metabolites and microbial communities and how those associations vary across environmental systems in future studies. By disentangling when, where, and how meta-metabolomes and microbial communities are driven by similar and divergent assembly processes, we will be able to incorporate the underlying driving abiotic and biotic forces into biogeochemical models.

## Discussion

Here we have presented a conceptual paradigm in which metabolite assemblages are treated as analogs to ecological communities and consider this to formalize the subdiscipline of 'meta-metabolome ecology.' While the analogy is not complete (e.g., metabolites do not evolve like biological species), there are numerous conceptual parallels that allow one to ask questions about environmental metabolites by applying ecology-inspired analyses to metabolite assemblages. We contend that the outcomes of these analyses provide deeper understanding of biochemical and biogeochemical dynamics. For example, identifying when, where, and why metabolite assemblages are governed by deterministic or stochastic processes offers information about key drivers that may be incorporated into predictive biogeochemical models. This presents exciting opportunities given the integral role of metabolites in emergent biogeochemical function[1,2,9]. For example, not all features of meta-metabolomes can or should be used in mechanistic models. Understanding assembly processes allows attention to be placed on the meta-metabolomes and parts of meta-metabolomes that are most important to capture in mechanistic models. Throughout the analyses discussed above, we have highlighted numerous insights that would have been challenging to uncover with traditional multivariate analyses (e.g., quantifying variation in stochastic and deterministic assembly processes impacting meta-metabolomes). We anticipate further development of the conceptual, theoretical, and methodological unification initiated here between meta-community ecology and metabolite assemblages.

While we focused heavily on dendrogram-based null models, there are other immediate opportunities to advance understanding by combining dendrogram-based and dendrogram-free null models. This approach has been pioneered in microbial ecology to parse out the relative influences of variable selection, homogeneous selection, dispersal limitation (combined with drift), and homogenizing dispersal (i.e., mass effects)[35,64,78,81]. More specifically, this is pursued by combining the βNTI null model with the identity-based Raup–Crick null model. Using this approach to refine the assembly processes influencing metabolite assemblages based upon the MCD revealed that variable selection, homogeneous selection, dispersal limitation (combined with drift), and homogenizing dispersal were responsible for 41.4%, 12.9%, 1%, and 39.5% of variation in pore and surface water meta-metabolomes, respectively. This demonstrates that just as in ecological systems, spatial processes can have significant influences over meta-metabolomes. To fully incorporate metabolites into predictive hydro-biogeochemical models, both selection-based and spatial processes will need to be considered.

Environmental metabolomics has significantly improved our understanding of ecosystem function and biogeochemical cycles[1,2,9]. By studying meta-metabolomes as analogs to meta-communities, we aim to deepen understanding of the factors influencing the spatial and temporal organization of metabolites within environmental systems. While our demonstration dataset was derived from a river corridor system, the conceptual and methodological framework used here can be applied to any system (e.g., soils, marine, human gut, etc.) and implemented with data collected using other high-resolution mass spectrometry

instruments (e.g., TOFMS, Obritrap-MS, IMS-MS). More importantly, this paradigm can be combined with various omics techniques (e.g., metagenomics, metatranscriptomics, metaproteomics) to broaden the relationships between meta-metabolome assembly and microbial activity. We anticipate further development of concepts, theory, and methods that deepen the analogy between metacommunities and meta-metabolomes, but that also diverge from this analogy when needed (e.g., metabolites are not competitively excluded). Application of the framework initiated here to a broad range of ecosystems will provide opportunities to elucidate key principles that are generalizable and that can be used to advance our capacity to predict system dynamics.

## Methods

**Sample collection.** River and pore water samples were collected from the Columbia River in Washington State along a ~1 km transect along the shoreline. Supplementary Fig. 1 shows the spatial location of these sites, which were along the mainstem of the river in southeast Washington State (Supplementary Table 1). In general, the field system is a gravel bedded river with a thin strip of riparian vegetation, though both of these features are variable across the sampling locations. The field site has been extensively studied and described in previous publications;[36,43,44] we point the reader to these previous publications for additional details. At each location, one replicate of river water was collected, and three pore water samples were collected and filtered using a 0.2 μm Sterivex filters (MilliporeSigma, MA, USA). Pore water replicates were collected from 30 cm depth within a 1 m$^2$ area using 0.25-inch diameter sampling tubes (MHE Products, MI, USA). Filters were stored at −80 °C until DNA could be extracted while water samples were stored at −20 °C until they could be used for further analysis.

**DNA extraction, sequencing, and processing.** DNA was extracted from Sterviex filters using a Powersoil DNA isolation kit (Mo Bio Laboratories, Inc., Carlsbad, CA). In order to generate 16 S rRNA gene data, the V4 region of 16 S rRNA genes was amplified and sequenced using the universal bacterial/archaeal primer set 515 F/806 R on an Illumina MiSeq instrument at Argonne National Laboratory according to the Earth Microbiome Project standard protocol[91]. Resulting 16 S rRNA amplicon sequences were analyzed using the open access 'hundo' pipeline[92]. Adapters were trimmed, low quality reads (i.e., length < 100 bp, quality score > 10), and contaminant sequences were filtered using BBDuk2 from the BBTools package[93]. Reads passing the quality filter were then clustered at 97% into de novo operational taxonomic units (OTUs) using VSEARCH with a minimum merge length of 150 bp and a minimum sequence abundance of 2[94]. Simultaneously, chimeric sequences were removed through de novo prediction and reference-based identification. Following clustering, BLAST was used to align sequences to the SILVA nr SSU reference database[95,96] and taxonomy was assigned based upon the CREST lowest common ancestor classifier[97]. Sequences were aligned using Clustal Omega[98] and a phylogenetic tree was generated using FastTree 2.0[99].

**FTICR-MS sample preparation and data collection/preprocessing.** Fourier Transform Ion Cyclotron Resonance Mass Spectrometry (FTICR-MS) was used for the ultrahigh resolution characterization of dissolved organic matter (DOM) within each sample. Filtered river and pore water samples were acidified to pH 2 with 85% phosphoric acid and extracted with PPL cartridges (Bond Elut), following Dittmar et al. (e.g., solid-phase extraction)[100]. High-resolution mass spectra of the DOM were collected using a 12 Tesla (12 T) Bruker SolariX Fourier transform ion cyclotron resonance mass spectrometer (Bruker, SolariX, Billerica, MA) located at the Environmental Molecular Sciences Laboratory in Richland, WA. Samples were directly injected into the instrument using a custom automated direct infusion cart that performed two offline blanks between each sample. The FTICR-MS was outfitted with a standard electrospray ionization (ESI) source, and data was acquired in negative mode with the needle voltage set to +4.4 kV, resolution was 220 K at 481.185 $m/z$. One hundred 44 scans were co-added for each sample and internally calibrated using organic matter homologous series separated by 14 Da (–CH$_2$ groups). The mass measurement accuracy was typically within 1 ppm for singly charged ions across a broad $m/z$ range (100–900 $m/z$). The FTMS peak picker module in the BrukerDaltonik Data Analysis software (version 4.2) was used to convert raw spectra to a list of $m/z$ values with a signal-to-noise ratio (S/N) threshold set to 7 and absolute intensity threshold to the default value of 100. Formularity[46], an in-house software, was used to assign chemical formulae following the Compound Identification Algorithm[45] and to align peaks with a 0.5 ppm threshold. Chemical formulae were assigned based on the following criteria following established methods: S/N > 7, and mass measurement error < 0.5 ppm, taking into consideration the presence of C, H, O, N, S, and P and excluding other elements[1,2,4,7].

The R package ftmsRanalysis[101] was used to remove peaks that either were outside the desired $m/z$ range (200–900 $m/z$) or had an isotopic signature, calculate derived statistics (Kendrick defect, double-bond equivalent, aromaticity index, nominal oxidation state of carbon, standard Gibb's Free Energy of carbon

oxidation), and organize the data[53–56]. Given that charge competition renders peak intensities less informative across systems[102], all analyses were conducted using binary presence/absence values rather than peak intensities with the absence of a peak defined as being below the limit of detection. In the end, 3162 of 22,029 peaks were assigned a molecular formula (Supplementary Fig. 3).

**Metabolite dendrogram estimation.** Three different metabolite dendrograms were generated: the molecular characteristics dendrogram (MCD), transformation-based dendrograms (TD), and transformation-weighted characteristics dendrogram (TWCD). Using the derived statistics calculated above (e.g., elemental composition, double-bond equivalents, modified aromaticity index, and Kendrick's defect), we can compare the potential molecular similarities between identified chemical formulae as in common classification studies[50–52]. Multivariate similarities were evaluated by measuring the Euclidean distance between chemical formulae (vegdist, vegan package v2.5-6)[103]. These Euclidean distances were then used to perform a UPGMA hierarchical cluster analysis (hclust, 'average' method, stats package). Given that the MCD requires assigned molecular formula, all 3162 assigned peaks were included with this method (Supplementary Fig. 3).

Unlike the MCD, the TD estimates molecular similarity by inferring potential biochemical transformations based upon ultrahigh mass resolution differences between identified metabolites[1,3,9,58,59]. For example, if the mass difference between two metabolites was 18.0343, that would putatively indicate a loss or gain of an ammonium group, while a mass difference of 103.0092 would putatively indicate loss or gain of a cysteine. This calculation is enabled by the ultrahigh mass resolution of FTICR-MS data; given this resolution, using a transformation database (Supplementary Data 4), we considered any between-metabolite mass difference within 1 ppm of the expected mass of a transformation to be a match. Using these pairwise mass differences and transformation associations, we generated a transformation network in which nodes represent individual metabolites and edges are identified transformations. Relationships between metabolites were determined by selecting the largest cluster of interconnected nodes (discarding any node outside this cluster) and measuring the stepwise distance between each pair of metabolites (i.e., the minimum number of transformations required to connect one metabolite to another metabolite within the largest cluster of the biochemical transformation network). These pairwise distances were then standardized between 0 and 1. A UPGMA hierarchical cluster analysis (hclust, 'average' method, stats package) was then used to convert these distances into a dendrogram. The lack of required molecular formula enables this method to access a broader range of peaks than the other two methods (15979), though few peaks still have molecular formula information (Supplementary Fig. 3).

The TWCD is a composite dendrogram requiring partial creation of both the MCD and TD. First, a Euclidean molecular characteristics distance matrix must be created based upon the elemental composition and derived statistics (i.e., double-bond equivalents, etc.). Next, the standardized stepwise transformation distance matrix (i.e., values between 0 and 1) must be generated based upon the transformation analysis described above. Using simple matrix multiplication, the molecular characteristics matrix is combined with the standardized transformation matrix. In effect, this results in a matrix where realized molecular characteristic differences are down-weighted. The TWCD is the generated by performing a UPGMA hierarchical clustering analysis on this transformation-weighted, molecular characteristics distance matrix. Given that not all peaks have both transformations and assigned molecular formula, the TWCD features the fewest number of incorporated peaks (2384; Supplementary Fig. 3)

**α- and β-diversity analyses.** Taxonomic richness was calculated by counting the total number of metabolites in each sample. Numerous dendrogram-based (i.e., phylogenetic) α-diversity measurements were also utilized within this study and are more completely detailed within Tucker et al. (2016). In brief, three dendrogram-based α-diversity metrics were utilized: Faith's Phylogenetic Diversity (PD; here, referred to as Dendrogram Diversity or DD)[47], mean pairwise diversity (MPD)[65], and mean nearest taxon distance (MNTD)[65]. Faith's PD (or DD) is a richness measurement that sums the total length of dendrogram branches that connect taxon:

$$\text{PD } or \text{ DD} = \sum_{b \in B} L_b$$

where $b$ is a given branch with a larger set of branches $B$ and $L_b$ is the length of branch b. MPD is a metric, which determines the average dendrogram distance between taxon:

$$\text{MPD} = \frac{\sum_{ij} d_{ij}}{S(S-1)}$$

where $d_{ij}$ is the distance between taxon $i$ and taxon $j$ and $S$ is the total number of species within the community. Lastly, MNTD is a metric, which determines the average dendrogram distance between nearest neighbors (i.e., the next closest taxon on the dendrogram):

$$\text{MNTD} = \frac{1}{S} \sum_i d_{i \min}$$

where $d_{i\,min}$ is the shortest distance from taxon $i$ to all other taxon. Faith's PD was calculated using the *pd* function in the picante R package (v1.8)[104] while MNTD and MPD were measured using the *generic.metrics* function in the pez R package (v1.2-0)[105]. These metrics were calculated using each of the three dendrograms.

Taxonomic β-diversity was visualized by generating a Jaccard dissimilarity-based principal coordinate analysis (PCoA; *pcoa*, ape package v5.3). Using each of the dendrograms, dendrogram-based (i.e., phylogenetic) beta-diversity was calculated using the unweighted UniFrac metric (*GUniFrac*, GUniFrac package v1.1)[106]. UniFrac results were visualized using a PCoA (*pcoa*, ape package v5.3)[107]. In order to determine whether or not significant differences existed between groups using both the Jaccard and UniFrac distances, we performed PERMANOVA analyses (*adonis*, vegan package v2.5-6).

**α- and β-diversity ecological null modeling**. Both α- and β-diversity based null modeling was performed throughout this study. In order to assess whether α-diversity was more or less structured than would be expected by random chance, both net relatedness index (NRI) and nearest taxon index (NTI) were calculated[22,23,65]. NRI is the null model variant of MPD and measures the degree of dispersal across the entire dendrogram, while NTI is the null model variant of MNTD and identifies tip-level clustering. For each of these null model calculations, 999 randomized metabolite assemblages were generated through tip shuffling. The finalized index was then calculated:

$$NRI = -1\left(\frac{MPD_{obs} - \overline{MPD_{null}}}{MPD_{sd}}\right)$$

$$NTI = -1\left(\frac{MNTD_{obs} - \overline{MNTD_{null}}}{MNTD_{sd}}\right)$$

where $MPD_{obs}$ and $MNTD_{obs}$ are the observed α-diversity values, $\overline{MPD_{null}}$ and $\overline{MNTD_{null}}$ are the average α-diversity values derived from the null assemblages, and $MPD_{sd}$ and $MNTD_{sd}$ are the standard deviations of α-diversity values from the null assemblages. Both NTI and NRI can be interpreted similarly; positive values suggest phylogenetic clustering while negative values indicate phylogenetic overdispersion[65].

β-diversity null modeling was performed to investigate whether metabolite assemblages were significantly more or less similar than would be expected by random chance alone and to assess whether assemblages were deterministically or stochastically assembled. To explore these ecological assembly processes, we first calculated the dendrogram-based β-nearest taxon index (βNTI) for each possible pairwise comparison according to 10. First, β-mean nearest taxon index (βMNTD) for the observed metabolite assemblages must be calculated in order to estimate dendrogram-based turnover:

$$\beta MTND = \frac{\sum_{i_k=1}^{n_k} f_{i_k} \min\left(d_{i_k j_m}\right) + \sum_{i_m=1}^{n_m} f_{i_m} \min\left(d_{i_m j_k}\right)}{2}$$

where $f_{i_k}$ is the relative abundance of metabolite $i$ in community $k$, $n_k$ is the number of metabolites in community $k$, and $\min\left(d_{i_k j_m}\right)$ is the minimum dendrogram distance between metabolite $i$ in community $k$ and metabolite $j$ in community $m$. This metric was calculated using the *comdistnt* function (abundance.weighted = FALSE) in the picante R package (v1.8)[104]. Similar to the NRI and NTI calculations, 999 randomized communities were generated by shuffling the tips of the dendrogram. βMNTD was then determined for each of these null communities and βNTI was calculated:

$$\beta NTI = -1\left(\frac{\beta MTND_{obs} - \overline{\beta MTND_{null}}}{\beta MTND_{sd}}\right)$$

where $\beta MNTD_{obs}$ is observed βMNTD for the observed assemblages, $\overline{\beta MTND_{null}}$ is the average βMNTD for the null communities, and $\beta MTND_{sd}$ is the standard deviation of $\beta MTND_{null}$ values.

In order to compare the different dendrogram estimation methods, we independently calculated βNTI values for each of the three metabolite dendrograms (e.g., MCD, TD, and TWCD). Microbial βNTI values were generated using the 16 S rRNA gene amplicon phylogenetic tree. We also calculated Resulting βNTI values help examine phylogenetic turnover across samples, providing insight into the ongoing deterministic and stochastic ecological assembly processes occurring within the system. If a |βNTI| value is greater than 2, deterministic processes explain the observed assemblage differences; if a |βNTI| value is less than 2, stochastic processes are responsible for assemblage differences. Stochastic processes are those which are more random in nature and typically arise from dispersal-based events. Deterministic processes are those which are driven by environmental filtering, pushing assemblages to either be more or less similar than expected by random chance. Deterministic processes could be further broken down into variable selection if βNTI is greater than 2, and homogenous selection if βNTI is less than −2. Variable selection occurs when the environment drives assemblages to be significantly divergent, as observed when distinct geochemistry supports different microbial communities[38]. In contrast, homogeneous selection occurs when some common pressure push communities toward a similar configuration as has been observed in microbial communities experiencing salt stress in a soil succession system[62]. Correlations between microbial

and metabolite βNTI values were performed by averaging the βNTI values for a given assemblage and relating them within a sample.

In addition to the dendrogram-based βNTI, we distinguished stochastic processes by using the identity-based Raup–Crick (RC)[24,35]. Using 9999 iterations per pairwise comparison, null communities were probabilistically generated based upon observed metabolite assemblages and presence/absence-based Bray–Curtis (i.e., Sørensen) dissimilarities were calculated. The null distribution of these dissimilarity values was then compared to the observed Bray–Curtis value in order to measure the deviation from the null expectation. These deviations were then standardized to vary between −1 and 1, resulting in the finalized RC metric. If a |RC| value is greater than 0.95, the turnover between the compared assemblages was the result of either dispersal limitation or homogenizing dispersal. Dispersal limitation (RC > 0.95) occurs when assemblages are unable to mix resulting in significant ecological drift. Conversely, homogeneous dispersal (RC < −0.95) occurs when environments drive substantial mixing resulting in assemblages that are more similar than random chance alone. However, if a |RC| value is less than 0.95, the assemblages were as different as would be expected by random chance because no single process is able to dominate (i.e., weak selection and weak dispersal). Under these 'undominated' circumstances, no single assembly process is capable of dominating. Significant differences in distributions of both βNTI and RC values across surface and pore water classifications were identified using Mann–Whitney U tests (*wilcox.test*, stats package).

**Metabolite activity comparisons**. In order to identify putatively more active metabolites, we leveraged potential biochemical transformation information collected during the generation of the transformation-based dendrogram (TD). Specifically, we counted the number of transformations associated with each individual metabolite and arbitrarily decided that metabolites involved in more than 40 transformations were considered 'active.' This number was selected because it was the first instance where only 1 peak had a given number of transformations associated with it (Supplementary Fig. 4). Conversely, metabolites involved in no transformations were considered 'inactive.' Given that our transformation database is not exhaustive and that our active cutoff is fairly arbitrary, we caution that these classifications are not generalizable but serve as a sufficient exercise to demonstrate the potential utility of our framework. Using these two metabolite subassemblages (e.g., active and inactive), we calculated βNTI for all three of the metabolite dendrograms by pruning the dendrograms to match the data.

**Plot generation**. All boxplots, pie charts, and standard bar charts were generated using ggplot2[108]. Dendrograms and their associated bar charts were visualized using ggtree[109].

**Reporting summary**. Further information on research design is available in the Nature Research Reporting Summary linked to this article.

## Data availability

Microbial 16 S rRNA gene sequencing data are accessible from NCBI via the Bioproject number PRJNA576070, Biosample numbers SAMN16450882 through SAMN16450904. Peak-picked, unaligned FTICR-MS data are accessible on ESS-DIVE at https://data.ess-dive.lbl.gov/view/doi:10.15485/1675028[110]. The aligned FTICR-MS data report used in this study is available on GitHub at https://github.com/danczakre/Meta-Metabolome_Ecology.

## Code availability

All scripts used throughout the manuscript and more generalizable scripts are available on GitHub at https://github.com/danczakre/Meta-Metabolome_Ecology.

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

## Acknowledgements
We thank Evan Arntzen and Doug McFarland for help in sample collection. Pacific Northwest National Laboratory is operated by Battelle Memorial Institute for the U.S. Department of Energy under Contract No. DE-AC05-76RL01830. This research was supported by the U.S. Department of Energy (DOE), Office of Biological and Environmental Research (BER), as part of BER's Subsurface Biogeochemistry Research Program (SBR). This contribution originates from the SBR Scientific Focus Area (SFA) at the Pacific Northwest National Laboratory (PNNL).

## Author contributions
R.E.D. and J.C.S. conceived the study and drafted the manuscript. A.E.G. and J.C.S. collected samples. R.E.D. performed the data analysis, wrote necessary scripts, and generated the figures. R.K.C. and J.T. generated the FTICR-MS data. S.J.F. generated the microbial sequence data. A.E.G., E.G.B., and M.M.T. provided consultation across the various data types and analyses. All authors contributed to editing the manuscript.

## Competing interests
The authors declare no competing interests.
