## [Peer Review File · Nature Communications]

Reviewers' Comments:

Reviewer #1:

Remarks to the Author:

The paper by Danczak et al. describes a novel paradigm for investigating the connection between ecological systems and their molecular processes. Using two complementary, and a third combined dendrogram methodologies, authors were able to utilize traditional ecological theory and statistical approaches to investigate analogous metabolic processes. Though the connection between ecological theory and metabolites is incomplete, unique insights into the deterministic and stochastic processes that structure the metabolome were investigated. Using the Columbia River system in Washington state as a proof-of-concept system, authors were able to identify that molecular properties were stochastic while metabolite transformations were deterministically structured. Additional insights into the biochemical activity of metabolites were further addressed using the novel system.

While the specific system studied led to some counter-intuitive results (microbial community did not affect metabolome assembly processes), previously unanswerable questions were addressed, and new questions made available upon further application. Additional use of the included analyses in other systems will likely occur and the pipeline may be optimized further as more uses are found. A discussion of the connection of meta-ecology and other omics is however missing and desired. Critiques to both the main text and supplemental are detailed below that warrant responses.

General text:

Line 69: remove "like"

Line 70: Either add "-MS" after orbitrap or combine FTCIR-MS and Orbitrap into single idea

Lines 69 – 71: Why are FTCIR- and Orbitrap-MS the only two high-resolution combinations mentioned? Other high-resolution systems have been utilized to investigate ecometabolomics. Expand concept for accuracy and add citations for FTCIR/Orbitrap/other HR-MS investigations.

Lines 93 – 95: Natural communities (not laboratory based) are also affected by human/anthropogenic means, specifically with regards to synthetic molecules/environmental pollutants or contaminants. The authors make clear in line 83 that the focus of the study is on NOM, however the effects of non-community derived metabolites is not addressed. Please comment on the predicted role of these types of molecules in structuring communities and how they are being treated in the current study.

Line 202: add "on" before "the Compound"

Line 240: add citation to other methods to generating metabolite dendrograms, or a comparison of methods if possible.

Line 260: change "that" to "than"

Line 301: Authors suggest that a common "conceptual grounding" allows for the development of overarching theory. Considering the range of environments cited in previous lines, this is an attractive aspect of the work, yet the statement currently has little support. Strengthen this with any "across system" investigations if possible.

Lines 404 – 408: Was a comparison of microbial consortia of pore water vs. river water conducted? Specifically, the statement that pore water contains a wider diversity of metabolic properties that are separated by more transformations – how does the analysis consider the relative abundance of microbial abundance in these different waters?

Lines 519 – 522: Is there a way to estimate the relative influence of these processes on the results? Specifically, can the influence of groundwater mixing (or other processes in unrelated systems) be estimated to predict their relative importance/strength in the current system/model?

Line 563: What is the rationale behind the cutoff of 40 or more transformations?

Lines 603 – 625: This data is required to fully understand 331-336. I suggest augmenting lines 331-336 so as not to introduce results/conclusions that are introduced much later. Additionally, this finding is quite interesting, as the authors mention. Are there other known studies/systems, likely marine/aquatic for comparability to this system, in which the microbial consortium is predicted or experimentally shown to influence the metabolome to a small degree, similar to this study? I do not mean to suggest additional studies exist that related the assemblage of microbial communities to metabolic assemblages as this study does, but simply systems in which the influence of the microbial loop is less consequential than other factors?

Line 590 – Similar to lines 404-408, where is the microbial assemblage data?

Figure 2a: Data never mentioned in main text or discussed. Please address.

SUPPLEMENTAL:

Discussions in the supplemental file regarding the pros/cons of each dendrogram were well appreciated. I however think these should have been addressed in the main text. For example, only 13% of the metabolites detected via FTICR-MS analysis were included in the MCD dendrogram. This point is important to make clearer in the main text for comparison of the models.

MCD dendrogram: The description of how related compounds cluster, with the example of sulfonate, was well received. However, with larger molecules, especially those with constitutional isomers, the relatedness due to composition, DBE, etc. may be overestimated. A brief discussion of the impact of this would be appreciated.

Sample collection: There is no mention of the number of sites where replicates were taken. Please add. A small description of the 1 km transect would also help as this is a meta-ecology study. What factors influence this region, how large is the watershed, is this heavily wooded, etc. A map of the area with watershed could also be included but is not needed if a site description is included.

FTICR-MS: molecular formulas were restricted to 0.5 ppm error; however, no mention is made of what occurred if/when a mass resulted in two candidate formula. Did this occur and if so, how was it corrected for? Additionally, why was this error used as the cutoff? If the error were for example raised to 1.0 ppm, what is the increase in the number of metabolites with formulas that could be included in the MCD dendrogram and further how does that affect the comparisons?

Additional table requested: It would be important to include a table with the number of spectral features detected, how many remained after processing, how many were assigned formula based on the above stated constraints, and then how many metabolites were included in each model (MCD vs TD vs TWCD). Could be made into two tables if more appealing as such.

Supplemental excel file: Some masses do not match the structures/compounds. For example, CH₄_O has a mass of 0.03638 in the associated mass column. This is obviously incorrect. Please double check the file and correct all issues.

Broad picture: The current study connects community ecology with metabolomics data. However, I see clear connections to transcriptomics and proteomics that are never discussed. The community composition and "state" of the system will clearly influence the number of transcripts and their

expression, which would downstream connect to the metabolome. I am interested to see if the authors envisage similar studies that also include other omics data and a discussion of this would be a welcome addition.

I look forward to the authors thoughtful responses to my critiques.

Sincerely,
Remington X. Poulin, PhD

Reviewer #2:

Remarks to the Author:

In this manuscript, Danczak and colleagues propose the application of concepts from the field of metacommunity ecology for the inference of the mechanisms that govern the diversity and turnover of metabolomes in environmental samples. They first present their proposed framework and relate it to the existing framework for inferring microbial community diversity and assembly processes. Then, they apply the proposed framework on an example dataset from the Columbia River corridor where they examine the diversity and turnover of river water and subsurface pore water metabolomes and microbiomes. Since phylogenetic trees are not applicable to metabolites (as the authors state multiple times), the authors replace the phylogenetic trees that are used in metacommunity ecology with dendrograms. They first build two types of dendrograms: one based on the molecular properties and one based on a "transformation network" of the compounds with the latter network inferred using the mass difference among two given metabolites and a biochemical transformation database. Then they also merge the two approaches and they create a third type of dendrogram by weighing the molecular properties dendrogram with the distances from the transformation-based dendrogram. To infer stochastic turnover processes, they also use the Raup-Crick metric based on the Bray-Curtis dissimilarity similarly to the metacommunity ecology framework (except that here they transform metabolite data to presence/absence). Using this approach, they study metabolome α - and β -diversity patterns in the example dataset, the former by deploying dendrogram-informed α -diversity metrics and the standardized-effect-size versions of those and the latter by combining dendrogram-informed β -diversity null modeling and the Raup-Crick metric. They perform these analyses using all three types of dendrograms (and at different spatial scales and putative "metabolite activity" classes for β -diversity) and they discuss their findings in the light of what each dendrogram represents. They finally compare the inferred assembly process patterns between metabolomes and microbiomes and find that these patterns are not the same.

The framework presented in this study is novel and the logical arguments explaining why metabolomes could be analyzed following concepts from metacommunity ecology are mostly justified. Indeed, analyzing metabolomes with such a framework could provide very useful insights regarding the drivers of biochemical processes in environmental settings. However, there are several critical aspects that differentiate metabolomes from microbiomes that were not taken into account and that make the application of such a framework questionable for metabolome datasets. In specific:

- 1) Maybe the most critical aspect that is not addressed by the authors is the temporal scale along which these kinds of analyses make sense. This is a very critical aspect of performing analyses such as the ones proposed in this manuscript, because as the authors also state (Lines 621-624), metabolite assemblages change faster than microbial assemblages. If microbial profiling via 16S rRNA amplicons provides community "snapshots", metabolite analyses with low temporal resolution (or no temporal resolution at all) provide "snapshots of a snapshot". Can we really base conclusions regarding drivers of metabolome assembly on such datasets? In microbial ecology, we have a fair amount of evidence on how fast microbial communities change in nature that has been accumulated over years of observations and measurements. What is the respective knowledge on the stability of metabolomes and, consequently, what is the temporal resolution of sampling effort that is required so that such

analyses make sense? I would see the proposed framework being applicable mostly for experimental manipulations examining patterns before/after some kind of treatment and with a sampling scheme requiring high temporal resolution that is justified by the expected changes in the metabolism(s) present in the monitored communities. In the example dataset there is no temporal resolution.

2) Another issue comes with the (inevitable) use of dendrograms instead of phylogenetic trees for the proposed metabolome analyses. The authors use a Molecular Characteristics Dendrogram (MCD), a Transformation-based Dendrogram (TD) and a Transformation-Weighted Characteristics Dendrogram (TWCD – a «hybrid» of the former two) for their analyses, in an effort to represent some kind of «metabolic similarity» among the compounds in analogy to the phylogenetic similarity represented by phylogenetic trees in microbiome analyses. I am not convinced that any of the three proposed dendrograms manages to represent that similarity accurately. The MCD by itself is «biologically ignorant» in the sense that it merely depicts the chemical similarity among the compounds based on euclidean distances calculated from the elemental composition and three more chemical indices. Thus, closely related compounds in the MCD do not necessarily have similar metabolic utility and vice versa. This issue is partly addressed using the TD, but here again the transformation distance is calculated from a list of known transformations and it is therefore uncertain if the transformations can actually occur in the specific samples. I do not see a way to be sure of that in the absence of metatranscriptomic/metaproteomic data with which sample-wise metabolic pathways can be reconstructed and proper, biologically-meaningful, compound distances can be calculated. Obviously, the TWDC does not solve this problem as well; It just incorporates the error introduced by TD-distances into the already questionable MCD dendrogram topology. Moreover, it is not mentioned anywhere if and how an equivalent of phylogenetic signal can be estimated from the proposed dendrograms. As the authors well know, this is very important when deciding which phylogeny/dendrogram-informed diversity metric to use. In the example dataset the authors use metrics that are sensitive to differences at different dendrogram depths (MPD and MNTD and their β - and ses- implementations) and the results differ sometimes depending on the metric. In the absence of an equivalent of phylogenetic signal, how can we decide which metric «tells the truth» ?

3) Performing the proposed analyses on subsets of metabolites based on putative biochemical activity (Lines 558 and on) can also be problematic. First, this assumes a kind of exclusive metabolism for specific elements and no interactions with other parts of the metabolism, which is rarely the case. Then, the «putatively more active» compounds are defined arbitrarily here as the ones «involved in more than 40 transformations». Could the authors provide any further justification on how they chose this number? For example by visual examination of the transformation network or by estimating some network property?

On a side note, the structure of the manuscript is also a bit loose and a clear separation of the different manuscript parts (other than the Conclusions part) is lacking. There is also some text repetition here and there and some result statements would benefit from a figure reference. In specific:

Line 260: «that» <- «than»

Line 262: maybe add «z-scores quantifying the deviation from...» before «null»

Lines 389-391: Please add the respective figure reference

Lines 401-403: Repetition

Lines 404-406: Please add the respective figure reference

Lines 428-429: Repetition

Lines 455-458: Only if the clusters are biologically-ecologically meaningful, but in a dataset like this it is hard to know.

Lines 467-469: As the authors well know, the choice of metric is based on where the phylogenetic signal lies within the tree. Since here a respective test is obviously not applicable, how can the choice of β MNTD be justified? (see the second main comment)

Lines 484-485: The same as above. How do we choose the appropriate model and how do we know

which model tells «the truth»?

Lines 507-509: Yes but this makes sense if the concept of metacommunity makes sense in the examined ecosystem parts, i.e., if the spatially distinct samples within it are connected by dispersal. Is this true in the example dataset? I can understand it for the river water but I wonder if this is also true for the pore water.

Lines 524-525: Obviously not significant to the point of causing homogenization?

Lines 530-533: Why would mixing affect only the MCD but not the TD? In the end what matters is the metabolome matrix that will tend to be more similar on average due to mixing irrespectively of the nature of the compounds (similarly to the effect of mass transfer events in metacommunity ecology).

Reviewer #3:

Remarks to the Author:

High-resolution mass spectrometric techniques are revolutionising our ability to understand biogeochemical cycles. The major advance of this paper is a framework to investigate the mechanisms underlying the spatiotemporal dynamics of metabolites by integrating concepts and tools developed in the study of biological communities. The authors of this study argue that ecological communities offer a strong conceptual analogue to metabolite assemblages, so the principles for studying them are directly transferable. I cannot agree enough! These points have been alluded to in previous studies (e.g. Mentges et al. 2017 *Front. Mar. Sci.*; Tanentzap et al. 2019 *PNAS*), but a unified synthesis is sorely needed. Therefore, the framework developed in this paper offers a major conceptual anchor and will be transformative for the immediate field and more widely in ecology and biogeosciences.

The authors demonstrate proof-of-concept for their framework in the context of predicting the processes governing the assembly of metabolite communities. They present two main analytical approaches to analysing assembly processes using molecular properties and biochemical transformations with either dendrogram or dendrogram-free null modelling. The advantages of these approaches, and the dendrogram specifically, are the ability to determine whether the communities (and their composition) arise simply because of chance or are different (e.g. more similar) than random, such as if they are structured deterministically. The specific results of these analyses, applied to a small dataset from the Columbia River corridor, are not to be taken literally and I am far less concerned about what inferences we can draw in this specific example or even if they are "correct". For me, this paper is much more about the concepts. That said, I would have liked a bit more specific discussion on why resolving metabolite assembly dynamics is important.

I thought the work was technically sound, but I struggled to follow all the details, especially the estimation of the transformation-based dendrogram (TD), and I worry about its reproducibility. The TD is the minimum number of steps a given metabolite must take to reach another metabolite in the biochemical network, calculated for all pairwise comparisons. Yet it is not apparent to me how available these data are beyond this specific study system because the approach requires a database of putative biochemical transformations. Presumably the database used in this study (Supplemental File 4) is specific to the authors' observations and not entirely transferrable in other contexts. So if someone wanted to calculate TD afresh, how would they go about doing it? That information seems to be glossed over in the text and Supplementary Information. Somewhat related, I thought NRI and NTI say very different things about where differences lie in the dendrograms (e.g. roots vs tips; Webb et al. 2002 *ARES*), but the distinction is not explicitly presented in the text nor why these differences might matter. There is some attempt to address this point on lines 397 to 399, but I'm not sure it is clear enough and I think the statement on line 398 would be equally true of clustering towards the root.

Related to this last point, I thought there were opportunities to improve the clarity and causal inference of the presented claims to strengthen the general conclusions and their impact. First, like biological taxa, individual metabolites will undergo fluctuations in production and degradation. The

authors claim this process is analogous to birth and death rates. But unlike biological taxa, these rates are much more intrinsically linked because the degradation process produces (i.e. gives birth) to new products. This is related to my comment on line 145 about "inheritance" of elemental composition. So I think the analogies between biological taxa and metabolite compounds can be strengthened in this vein. Second, the authors claim metabolites are governed by passive movement on line 128. But are they? Sure, metabolites themselves can't move. But their vectors can and do so in relation to other metabolite gradients. How would these interactions influence the framework? Finally, the authors state that "opportunities to generate conceptual outcomes that would not be possible with traditional multivariate analyses". They then conclude that their analyses employing these methods demonstrate "that pore water metabolites (1) span a broader range of molecular properties and (2) are separated by a larger number of biochemical transformations." Won't "traditional multivariate analyses" reach these same conclusions? So I'm not sure this point about comparisons is what the authors really want to emphasise here. It seems like a false dichotomy.

A pedantic point that you should feel free to ignore is that I the meta-meta naming is pretty brain racking and unlikely to win over a general audience. I think it is etymologically sound, since a metabolome is an assemblage of metabolites from a sample and you have multiple assemblages in the environment, hence the "meta". And a similarity in definition can be drawn to genomics, where we have meta-genomics being the collection of genetic material from multiple environmental samples. But I think it gets somewhat muddled where metabolomes are defined on line 98 as the assemblages of metabolites from an ecosystem. So if metabolites are the assemblage, why do you need a meta? What does that differentiate? Assemblage of assemblages? I think if you are going to invent a word - which is fine - it just needs to be clearly defined. A further example of my confusion is on line 251 where we have "metabolome meta-assemblages". How is the meaning of this different from "metabolome assemblages". Both phrases mean groups of assemblages? The authors even conclude the paper by saying "metabolite assemblages are treated as analogs to ecological communities", so then why do you need the "meta"?

In sum, this is a terrific and inspirational foundation paper. I hope the authors see my comments as a reflection of my enthusiasm and desire to the broaden the uptake of their important conceptual work.
Andrew J. Tanentzap
ajt65@cam.ac.uk

Specific comments:

Line 69: Either "such as" or "like".

Line 101: Share should be plural.

Line 145: But they do pass on traits. They "evolve" as they are transformed through degradation processes and even pass on some of their properties, e.g. degradation products with P atoms will only have them if the parent compounds had a P atom, no? I think this point is also relevant on line 643.

Line 260: Replace "that" with "than"

Line 264: It would be more useful here to have the biological explanation and the results phrased in terms of stochastic vs deterministic structuring.

Line 321: I struggled with this prediction as you can have stochastic and deterministic processes being important, but your analyses will only detect one of these for a single dataset and it is unclear which. I also think this prediction needs to be better justified. Why would a distinct environment lead to deterministic signatures per se? Also, you seem to be conflating assembly processes within communities here (e.g. deterministic signatures) with between-site differences (i.e. "we expect ... river water and pore water ... will lead to"). The concept of "variable selection" also wasn't clearly enough defined for me in biological terms at this point in the text.

Line 334: I think you need to reconsider whether it is appropriate to reference the results here when you haven't presented them yet. I'm actually not overly concerned about the sample size - because like the authors say, the data are used for "demonstration purposes" and this paragraph could be moved later in the paper where the implications of the demonstration are discussed.

Line 341: I think it would be appropriate to recognise here that there are studies that employ relational information to determine whether differences exist between samples like Mentges et al. 2017 Front. Mar. Sci; Tanentzap et al. 2019 PNAS.

Line 372: And what advances in biogeochemistry would answering these questions deliver? I think you need to add some text to make this benefit explicit.

Line 449: The description of the statistical analysis is incorrect here. NMDS is an ordination technique for dimensionality reduction and thus visualising differences in multivariate data. It cannot produce "Pseudo-F" values. I suspect the authors used something like a permutational analysis of variance here (PERMANOVA), but these details, along with the performance of the NMDS, do not appear in the main text or supplementary information.

Line 517: But were these values significantly different from that expected by random, e.g. +2? Isn't that what matters here?

Line 528: Should you cite Figure 5b here?

Line 563: Please provide some evidence that 40 is a large number in your dataset, e.g. a histogram of number of transformations per molecule would show this point.

Line 566: Should processes be plural here to be grammatically consistent with vary?

Line 599: Why would you expect this biologically? The idea needs fleshing out.

Line 610: The conclusion for a "weak" association should come from the r value and not the p-values. P-values are either significant or not. There is no "weak" option as they shouldn't be interpreted as effect sizes. P-values are also biased by sample size in correlational tests. So you should ask if $r = 0.223$ is any less "weak" than $r = 0.298$... Who knows? And why dredge the data with both tests. Pick the most appropriate - likely the Spearman one given the small sample size.

Line 613: It isn't "contrary" to your hypothesis per se. In a Popperian framework, failure to reject the null doesn't mean you accept it. Maybe you can't reject the null because you haven't measured the right things (like suggested on lines 621 onwards) or the sample sizes are small or the characterisation of the assembly processes is incomplete.

Line 624: Or relationships with functional diversity as we have done previously (see Tanentzap et al. 2019 PNAS). Of course, perhaps the most parsimonious explanation is that you haven't measured the right things. Does that point need to be countered?

Line 638: But why does any of this matter? I think a general audience would appreciate some more specific exploration of why understanding coordination among assembly processes matters beyond being a fascinating aspect of biology.

Line 659: It help if the authors cited some of this "pioneering" work here.

Line 674: Why not just communities?

Line 683: I think you are missing a word between "that" and "be". Furthermore, this point about disturbance kind of comes out of nowhere. Disturbance isn't mentioned anywhere in the paper yet somehow appears in the final take-home message? I would have thought the point is more general than disturbance, i.e. you can predict system responses full stop.

Line 899: You should define acronyms like AI_mod here.

Line 907: Delete and just say something like "There were no significant differences between surface water and pore water samples as determined by a Mann Whitney U test." I also think this caption must explain what the boxes, vertical/horizontal lines, and points represent.

Line 922: Please specify the significance of the red dotted lines so the caption is self-contained. It might also help to specify that the p-values are for comparisons between pore-waters.

Figure 1: I think for clarity it would help if you could connect the MCD and TWCD approaches by numbering the metabolite structures on the far left in a way that would be consistent with the network presented below. Also, it would help if you specified if the numbered metabolites are the only ones measured, whereas the empty circles are assumed but unmeasured intermediates. Finally, I think the TWCD path is a bit black box. Any way to summarise graphically how you combine the TD and MCD data?

Figure 3: I find it surprising there is no significant difference and thus p-value reported for the top panel in b) which looks very similar to bottom panel in b) and bottom in panel in a), which oddly seem to have identical p-values (presumably because of the permutational nature of the analyses with the small sample size)?

Figure 4: The only place the PCoA acronym appears in the whole paper is in this figure and its associated caption. Not only does it need to be defined but why are you mixing PCoA with NMDS? The text needs to give some gentle explanation here as it is not immediately obvious. The SI doesn't even mention why this is done.

REVIEWER COMMENTS

Reviewer #1 (Remarks to the Author):

The paper by Danczak et al. describes a novel paradigm for investigating the connection between ecological systems and their molecular processes. Using two complementary, and a third combined dendrogram methodologies, authors were able to utilize traditional ecological theory and statistical approaches to investigate analogous metabolic processes. Though the connection between ecological theory and metabolites is incomplete, unique insights into the deterministic and stochastic processes that structure the metabolome were investigated. Using the Columbia River system in Washington state as a proof-of-concept system, authors were able to identify that molecular properties were stochastic while metabolite transformations were deterministically structured. Additional insights into the biochemical activity of metabolites were further addressed using the novel system.

While the specific system studied led to some counter-intuitive results (microbial community did not affect metabolome assembly processes), previously unanswerable questions were addressed, and new questions made available upon further application. Additional use of the included analyses in other systems will likely occur and the pipeline may be optimized further as more uses are found. A discussion of the connection of meta-ecology and other omics is however missing and desired. Critiques to both the main text and supplemental are detailed below that warrant responses.

General text:

Line 69: remove “like”

This extra word has been removed.

Line 70: Either add “-MS” after orbitrap or combine FTCIR-MS and Orbitrap into single idea

We’ve rearranged the sentence to attempt to improve wording

Lines 69 – 71: Why are FTCIR- and Orbitrap-MS the only two high-resolution combinations mentioned? Other high-resolution systems have been utilized to investigate ecometabolomics. Expand concept for accuracy and add citations for FTCIR/Orbitrap/other HR-MS investigations.

FTICR- and Orbitrap-MS are two of the most commonly used HR-MS techniques used in investigating complex mixtures in environmental settings (with Orbitrap-MS being more readily available). However, we recognize that there are more types of analysis and have expanded the sentence and added citations.

Lines 93 – 95: Natural communities (not laboratory based) are also affected by human/anthropogenic means, specifically with regards to synthetic molecules/environmental pollutants or contaminants. The authors make clear in line 83 that the focus of the study is on NOM, however the effects of non-community derived metabolites is not addressed. Please

comment on the predicted role of these types of molecules in structuring communities and how they are being treated in the current study.

Anthropogenic effects definitely contribute to metabolite assemblage turnover. To our mind, these impacts would be akin to dispersal-based processes, i.e., the “stochastic” introduction of new assemblage members to an existing NOM assemblage. Their persistence within the assemblage would then be governed by a combination of selection (i.e., degradability or persistence) or dispersal. Given that we are performing an untargeted survey of all molecular formula (due to the nature of our instrumentation), we are sampling those components of anthropogenic origin (though, admittedly, they will likely be hard to identify using FTICR-MS due to the lack of structural resolution). We point out this perspective briefly near the end of the first paragraph in sub-section “Conceptual parallels between metabolites and ecological units.”

Line 202: add “on” before “the Compound”

The word “on” has been added.

Line 240: add citation to other methods to generating metabolite dendrograms, or a comparison of methods if possible.

As far as we are aware, we are the first to generate metabolite dendrograms. Therefore, citations can't be provided. That being said, trait-based data have been used to generate dendrograms in plant communities in the past, and some example citations are included (e.g., Swenson et al., 2012 - Ecology).

Line 260: change “that” to “than”

“That” has been changed to “than”

Line 301: Authors suggest that a common “conceptual grounding” allows for the development of overarching theory. Considering the range of environments cited in previous lines, this is an attractive aspect of the work, yet the statement currently has little support. Strengthen this with any “across system” investigations if possible.

The conceptual grounding to which we were referring encompasses the utilization of similar bNTI and Raup-Crick methods across many different sites that feature many different questions. The purpose of this manuscript is to provide the same conceptual grounding to metabolomics; therefore, we are not certain what additional support we could provide specifically for metabolomics. However, cross-system studies exist that draw knowledge from applying null models to microbial communities. We have added citations for two such studies (Wang et al, 2013; Tripathi et al., 2018).

Lines 404 – 408: Was a comparison of microbial consortia of pore water vs. river water conducted? Specifically, the statement that pore water contains a wider diversity of metabolic properties that are separated by more transformations – how does the analysis consider the relative abundance of microbial abundance in these different waters?

This analysis does not consider relative microbial abundance as this particular analysis was focused on understanding which processes impact the meta-metabolome. The biochemical transformations that we identify are identified independently of any microbial information because many of them could be catalyzed by other organisms (or even abiotically) (see the references by Tfaily et al., Graham et al., Brietling et al., and/or Stegen et al. for a little more expansion on this approach).

Lines 519 – 522: Is there a way to estimate the relative influence of these processes on the results? Specifically, can the influence of groundwater mixing (or other processes in unrelated systems) be estimated to predict their relative importance/strength in the current system/model?

In theory, yes – that is very possible and microbially-based null modeling studies have utilized this information to estimate influences of mixing. Unfortunately, we do not have the necessary data to estimate mixing in this particular study as that would ideally be done with time series data. However, we note that the primary goal of our current manuscript is offering new concepts and methods as a template for future studies that dig into details of particular systems.

Line 563: What is the rationale behind the cutoff of 40 or more transformations?

The cut-off of 40 was based on an examination of a histogram of transformations; there was a substantial drop-off of number of transformations above 40. We have modified the language in order to convey this approach.

Lines 603 – 625: This data is required to fully understand 331-336. I suggest augmenting lines 331-336 so as not to introduce results/conclusions that are introduced much later. Additionally, this finding is quite interesting, as the authors mention. Are there other known studies/systems, likely marine/aquatic for comparability to this system, in which the microbial consortium is predicted or experimentally shown to influence the metabolome to a small degree, similar to this study? I do not mean to suggest additional studies exist that related the assemblage of microbial communities to metabolic assemblages as this study does, but simply systems in which the influence of the microbial loop is less consequential than other factors?

We think that our language may have led to some confusion regarding what this data indicates. We are not suggesting that the microbial community is not influencing the meta-metabolome. Instead, our results suggest that the environmental parameters which impact the metabolome assemblages might be divergent from those that impact the microbial community. This makes some sense given that metabolites face a whole suite of “selective” processes that don’t impact microbes in a similar way. For example, metabolites can be directly and near-spontaneously modified via photo-oxidation; microbes interface with light stress differently (i.e., phototrophy, UV resistance, etc.). We have adjusted the language in the following paragraph in an attempt to better convey our message. In addition, we removed mention of the results in material that was originally in lines 331-336, and now in the first paragraph of the section ‘Example use of the dendrogram-based framework.’

Line 590 – Similar to lines 404-408, where is the microbial assemblage data?

This section presents a number of results on the microbial communities, and their relationship to the metabolite assemblages. If we are missing something, we apologize. If the reviewer could elaborate on what they were hoping to see, that would be very helpful. If the reviewer is asking for additional analyses and data to be presented, we note that the microbial component of this work is intentionally less detailed than the metabolite analyses. This allows the manuscript to focus on the metabolites, but also point to how they can be connected to microbial communities through null model analyses. The manuscript is already very long, and we would like to avoid having it grow even longer unless there is a compelling need to include additional microbial data/analyses.

Figure 2a: Data never mentioned in main text or discussed. Please address.

We added a sentence describing the number of molecular formula, using Figure 2a as a reference.

SUPPLEMENTAL:

Discussions in the supplemental file regarding the pros/cons of each dendrogram were well appreciate. I however think these should have been addressed in the main text. For example, only 13% of the metabolites detected via FTICR-MS analysis were included in the MCD dendrogram. This point is important to make clearer in the main text for comparison of the models.

Only 13% (~14% after noticing a typo) of the compounds detected via FTICR-MS were capable of being assigned a formula; all molecular formulae were included on the MCD. This is unfortunately a deficiency with FTICR-MS formula assignment, rather than a deficiency of our methodology; this actually highlights where our methodology excels – the TD doesn't require formula assignment. The TWCD does have fewer included metabolites, however, as not all assigned formula have transformations. We have clarified this within the main text and added some more information about differences into the Supplementary Material.

MCD dendrogram: The description of how related compounds cluster, with the example of sulfonate, was well received. However, with larger molecules, especially those with constitutional isomers, the relatedness due to composition, DBE, etc. may be overestimated. A brief discussion of the impact of this would be appreciated.

This is an interesting perspective that probably can't be addressed using FTICR-MS data alone. The advantage of the MCD is that it takes a fairly agnostic approach in analyzing similarities between molecular formula; these relationships are intuitively related to potential chemical patterns. Using alternative methods like IMS-MS, GC-MS, LC-MS, or other alternative mass spectrometry approaches, we can gain more structural and biological information about each molecular formula. With this information, we can build modified dendrograms that will enhance the accuracy of the relationships.

Sample collection: There is no mention of the number of sites where replicates were taken. Please add. A small description of the 1 km transect would also help as this is a meta-ecology

study. What factors influence this region, how large is the watershed, is this heavily wooded, etc. A map of the area with watershed could also be included but is not needed if a site description is included.

Thank you for the suggestions, we have included additional details and a new supplementary figure (Supplementary Figure 1) that shows locations of the sites, and a new Supplementary File that has latitude and longitude for the sites. We also point readers to additional literature that has described the site in greater detail.

FTICR-MS: molecular formulas were restricted to 0.5 ppm error; however, no mention is made of what occurred if/when a mass resulted in two candidate formula. Did this occur and if so, how was it corrected for? Additionally, why was this error used as the cutoff? If the error were for example raised to 1.0 ppm, what is the increase in the number of metabolites with formulas that could be included in the MCD dendrogram and further how does that affect the comparisons?

The program that we use to assign molecular formula (Formularity; Tolic et al., 2017) has its own mechanisms for selecting the most likely formula if multiple could be assigned. Given a substantial body of literature, our analysis methodology (error rate, alignment stringency, elemental composition, S/N, etc.) is fairly standard and we did not perform any additional corrections (Tfaily et al., 2018, Tolic et al., 2017, Graham et al., 2017, 2018, Stegen et al., 2016, 2018). We have modified the “FTICR-MS Sample Preparation” subsection in the supplement to specify this.

While it is likely that the number of formulae would increase if the error was set to 1 ppm, we do not want to mention this because this might mislead readers down an incorrect analysis path. On a secondary note, the MCD will improve as formula assignment algorithms improve.

Additional table requested: It would be important to include a table with the number of spectral features detected, how many remained after processing, how many were assigned formula based on the above stated constraints, and then how many metabolites were included in each model (MCD vs TD vs TWCD). Could be made into two tables if more appealing as such.

This is a good idea. Instead of a table, however, we have added a new supplemental figure detailing the number of peaks detected across the dataset, assigned a formula, and incorporated into each of the dendrograms.

Supplemental excel file: Some masses do not match the structures/compounds. For example, CH4_O has a mass of 0.03638 in the associated mass column. This is obviously incorrect. Please double check the file and correct all issues.

This is actually not incorrect – this particular transformation is a substitution between a CH4 group and an oxygen. It is more of an issue with the label that’s being used. We have clarified the nomenclature in the Table legend.

Broad picture: The current study connects community ecology with metabolomics data. However, I see clear connections to transcriptomics and proteomics that are never discussed. The

community composition and “state” of the system will clearly influence the number of transcripts and their expression, which would downstream connect to the metabolome. I am interested to see if the authors envisage similar studies that also include other omics data and a discussion of this would be a welcome addition.

Yes, we definitely envisage an application of our methods to other ‘omics data and thank you for pointing out the omission of associated discussion of these potential applications. We added text throughout the manuscript to discuss how other ‘omics techniques can be used to improve ecosystem level conclusions when partnered to meta-ecosystem metabolomics.

I look forward to the authors thoughtful responses to my critiques.

Sincerely,
Remington X. Poulin, PhD

Reviewer #2 (Remarks to the Author):

In this manuscript, Danczak and colleagues propose the application of concepts from the field of metacommunity ecology for the inference of the mechanisms that govern the diversity and turnover of metabolomes in environmental samples. They first present their proposed framework and relate it to the existing framework for inferring microbial community diversity and assembly processes. Then, they apply the proposed framework on an example dataset from the Columbia River corridor where they examine the diversity and turnover of river water and subsurface pore water metabolomes and microbiomes. Since phylogenetic trees are not applicable to metabolites (as the authors state multiple times), the authors replace the phylogenetic trees that are used in metacommunity ecology with dendrograms. They first build two types of dendrograms: one based on the molecular properties and one based on a “transformation network” of the compounds with the latter network inferred using the mass difference among two given metabolites and a biochemical transformation database. Then they also merge the two approaches and they create a third type of dendrogram by weighing the molecular properties dendrogram with the distances from the transformation-based dendrogram. To infer stochastic turnover processes, they also use the Raup-Crick metric based on the Bray-Curtis dissimilarity similarly to the metacommunity ecology framework (except that here they transform metabolite data to presence/absence). Using this approach, they study metabolome α - and β - diversity patterns in the example dataset, the former by deploying dendrogram-informed α -diversity metrics and the standardized-effect-size versions of those and the latter by combining dendrogram-informed β -diversity null modeling and the Raup-Crick metric. They perform these analyses using all three types of dendrograms (and at different spatial scales and putative “metabolite activity” classes for β -diversity) and they discuss their findings in the light of what each dendrogram represents. They finally compare the inferred assembly process patterns between metabolomes and microbiomes and find that these patterns are not the same.

The framework presented in this study is novel and the logical arguments explaining why metabolomes could be analyzed following concepts from metacommunity ecology are mostly justified. Indeed, analyzing metabolomes with such a framework could provide very useful

insights regarding the drivers of biochemical processes in environmental settings. However, there are several critical aspects that differentiate metabolomes from microbiomes that were not taken into account and that make the application of such a framework questionable for metabolome datasets. In specific:

1) Maybe the most critical aspect that is not addressed by the authors is the temporal scale along which these kinds of analyses make sense. This is a very critical aspect of performing analyses such as the ones proposed in this manuscript, because as the authors also state (Lines 621-624), metabolite assemblages change faster than microbial assemblages. If microbial profiling via 16S rRNA amplicons provides community “snapshots”, metabolite analyses with low temporal resolution (or no temporal resolution at all) provide “snapshots of a snapshot”. Can we really base conclusions regarding drivers of metabolome assembly on such datasets? In microbial ecology, we have a fair amount of evidence on how fast microbial communities change in nature that has been accumulated over years of observations and measurements. What is the respective knowledge on the stability of metabolomes and, consequently, what is the temporal resolution of sampling effort that is required so that such analyses make sense? I would see the proposed framework being applicable mostly for experimental manipulations examining patterns before/after some kind of treatment and with a sampling scheme requiring high temporal resolution that is justified by the expected changes in the metabolism(s) present in the monitored communities. In the example dataset there is no temporal resolution.

We suggest that our framework is well suited for studying these sorts of temporal dynamics and that they do not need to be understood *a priori*. While it is certain that molecular formula pools can change more rapidly than microbial communities, we would argue that the temporal scales of these dynamics are not that different from transcript pools. As with functional profiles delineated from sequenced transcripts, there is reason to believe that there are ‘steady state’ like conditions that exist in assemblages of molecular formula. For example, Stegen et al., 2018; Graham et al., 2017; Graham et al. 2018, all suggest that there are “characteristic” molecular formulae in pore water/surface water end members. Such characteristic behavior suggests that, regardless of time scale, there are certain formula that are always present. Other studies (i.e., Dalcin Martins et al., 2017, Danczak et al., pre-print) using NMDS plots have shown that there are temporally consistent patterns as well. Unfortunately, the direct mechanism impacting temporally consistent or divergent molecular formula composition isn’t as well studied. It is our hope that our framework can start attaching a process through which similarities and divergences arise.

In addition, the null model analyses are inherently tied to the spatial and temporal scales of sampling/analysis. This is a powerful feature of null models as it provides an opportunity to study systems across scales using a consistent conceptual and analytical framework. Discussion of scale dependence is provided in sub-section “Changing scales leads to additional insights accessible only through null modeling based on relational information.” This further indicates that even for systems that are not in steady state, the framework and associated analyses/outcomes provide a window through which to understand meta-metabolomes that is both useful and novel.

2) Another issue comes with the (inevitable) use of dendrograms instead of phylogenetic trees for the proposed metabolome analyses. The authors use a Molecular Characteristics Dendrogram

(MCD), a Transformation-based Dendrogram (TD) and a Transformation-Weighted Characteristics Dendrogram (TWCD – a «hybrid» of the former two) for their analyses, in an effort to represent some kind of «metabolic similarity» among the compounds in analogy to the phylogenetic similarity represented by phylogenetic trees in microbiome analyses. I am not convinced that any of the three proposed dendrograms manages to represent that similarity accurately. The MCD by itself is «biologically ignorant» in the sense that it merely depicts the chemical similarity among the compounds based on euclidean distances calculated from the elemental composition and three more chemical indices. Thus, closely related compounds in the MCD do not necessarily have similar metabolic utility and vice versa. This issue is partly addressed using the TD, but here again the transformation distance is calculated from a list of known transformations and it is therefore uncertain if the transformations can actually occur in the specific samples. I do not see a way to be sure of that in the absence of metatranscriptomic/metaproteomic data with which sample-wise metabolic pathways can be reconstructed and proper, biologically-meaningful, compound distances can be calculated. Obviously, the TWDC does not solve this problem as well; It just incorporates the error introduced by TD-distances into the already questionable MCD dendrogram topology. Moreover, it is not mentioned anywhere if and how an equivalent of phylogenetic signal can be estimated from the proposed dendrograms. As the authors well know, this is very important when deciding which phylogeny/dendrogram-informed diversity metric to use. In the example dataset the authors use metrics that are sensitive to differences at different dendrogram depths (MPD and MNTD and their β - and ses- implementations) and the results differ sometimes depending on the metric. In the absence of an equivalent of phylogenetic signal, how can we decide which metric «tells the truth» ?

We do not suggest that any of our dendrograms represent a ‘metabolic similarity’ but rather they represent similarities between molecular formula (MCD), biochemical transformations (TD), and an interplay of biochemical transformations and molecular formula (TWCD). This framework does indeed lead to a ‘biologically ignorant’ point of view – we view this as one of the key strengths of our methodology. One key consideration with FTICR-MS-derived information is that it captures any ionizable molecular formula, not just those that are biologically active. Past studies have demonstrated that, despite this potential limitation, assemblages of molecular formula still conveys biogeochemically relevant information (i.e., Graham et al, 2018 - STOTEN, Stegen et al, 2018 – Nat Comm, Garayburu-Caruso et al, 2020 – EST Lett, etc.). The MCD allows us to investigate which assembly processes may impact the entire body of these molecular formula, rather than making inaccurate assumptions regarding biological activity. Such relationships have allowed others in the past to derive a “functional diversity” metric for FTICR-MS derived molecular formula as well (Mentges et al, 2017 – Front Mar Sci, Tanentzap et al, 2019 – PNAS).

As alluded to, the TD enables one to narrow the metabolites to those that are putatively biochemically active (and thereby more biologically relevant) and gain further insight into potential biochemical interactions. The underlying transformation analysis used to obtain the TD has demonstrated a strong capability to capture biogeochemical dynamics in river corridors (Graham et al., 2018 - STOTEN). However, it is important to note that many of these transformations can be catalyzed through abiotic mechanisms as well (i.e., photoactivity, abiotic oxidation interactions, etc.).

There is inherent value in investigating the processes which give rise to all molecular formula, not just those involved in microbiologically mediated reactions. These molecular formulae come from and interact with many aspects of the environment (e.g., plants, fungi, humans, microbes, geochemical processes). We argue that there is no way (yet) to know which molecules may or may not be relevant to the functioning of any given system. As such, we believe that studying organic matter in an untargeted manner provides us with information inaccessible to traditional ‘omics techniques. We do not believe that these limitations hinder the ability of each dendrogram to capture assembly, but rather highlight an area where complementary data (i.e., metaG, metaT, metaP) will improve our understanding of biogeochemistry. We have added some text within the “Building metabolite dendrograms” section to clarify this viewpoint.

Regarding the necessity for a phylogenetic signal, we argue that it is not necessary for the dendrograms given that our framework is based on functional trait-like data. A phylogenetic signal is certainly necessary when using a phylogenetic tree to estimate the degree of trait-relatedness between community members. While it is measured differently within microbial communities, the principle is the same: how well are niches (i.e., traits) conserved across phylogenetic distances. Similar to Swenson, Stegen et al, 2011 – Ecology, a phylogenetic signal is unnecessary when using molecular properties or transformations because these are the traits themselves. As far as which dendrogram tells the truth, each dendrogram provides a different information, all of which is valid. For example, the bNTI results for a microbial community may vary substantially when comparing results from a *dsrAB* tree and *narG* tree, but neither is less true. It is the onus of the researchers to explain what divergences mean when and where they arise. We have expanded upon this point of view in a new section titled “A quick note about phylogenetic signals.”

3) Performing the proposed analyses on subsets of metabolites based on putative biochemical activity (Lines 558 and on) can also be problematic. First, this assumes a kind of exclusive metabolism for specific elements and no interactions with other parts of the metabolism, which is rarely the case. Then, the «putatively more active» compounds are defined arbitrarily here as the ones «involved in more than 40 transformations». Could the authors provide any further justification on how they chose this number? For example by visual examination of the transformation network or by estimating some network property?

The cutoff of 40 transformations is fairly arbitrary. It was based upon the visual examination of a transformation count histogram (Supplementary Figure 4). Based upon this histogram, there were no metabolites associated with 40 transformations (the first time this happened on the dendrogram). Based upon this, we decided that metabolites with more than 40 transformations could be considered “putatively more active” than those involved in no transformations. It is important to note that given our transformation analysis is database limited, there are likely underestimates of the number of transformations a given metabolite is putatively involved in. However, our purpose for this analysis was not to parse out activity vs. non-activity as our data are insufficient to do so, but rather give an example of how our framework can produce conceptually new conclusions. While it would be exciting to see our conclusion that putatively more active peaks are more likely to experience variable selection maintained across studies, we would be equally excited to see different outcomes in future studies using our methodology. In

the manuscript we have clarified why 40 transformations was selected, expanded that section slightly, clarified that the choice was somewhat arbitrary, and added the histogram as Figure S4.

On a side note, the structure of the manuscript is also a bit loose and a clear separation of the different manuscript parts (other than the Conclusions part) is lacking. There is also some text repetition here and there and some result statements would benefit from a figure reference.

While we do have a non-traditional manuscript format, that is due to our manuscript conveying conceptual theory, rather than a specific experimental design. However, we tried to organize the manuscript as logically as possible with many headers and sub-headers to guide the reader.

In specific:

Line 260: «that» <- «than»

We have changed 'that' to 'than'

Line 262: maybe add «z-scores quantifying the deviation from...» before «null»

Added.

Lines 389-391: Please add the respective figure reference

We have added a reference to Figure 3.

Lines 401-403: Repetition

This sentence is intentionally repetitious with earlier sentences in order to cue readers to the upcoming paragraph.

Lines 404-406: Please add the respective figure reference

We have added a reference to Figure 3 at the end of the sentence.

Lines 428-429: Repetition

As with lines 401-403, this sentence is intentionally repetitious.

Lines 455-458: Only if the clusters are biologically-ecologically meaningful, but in a dataset like this it is hard to know.

Regardless of the meaning behind the clusters, they would not have been uncovered without the utilization of a dendrogram-resolved approach. It is our hope that when such patterns arise in other datasets, researchers will take the necessary care to explain the clustering results.

Lines 467-469: As the authors well know, the choice of metric is based on where the phylogenetic signal lies within the tree. Since here a respective test is obviously not applicable, how can the choice of β MNTD be justified? (see the second main comment)

Given that a phylogenetic signal isn't necessary for trait-based dendrograms, we selected bNTI due to substantial literature supporting its interpretation (e.g., Chase et al., 2011, Stegen et al., 2012, 2013, 2015, 2016, Tucker et al., 2016, Zhou and Ning, 2017, Danczak et al., 2020). However, we also believe that the utilization of trait-based dendrograms allows for other null models to provide additional information. We look forward to additional work that uses multiple null modeling methods in complementary fashion to gain deeper insights than any one approach can provide. The associated text can be found in the 4th paragraph of sub-section "Spatiotemporal variation in molecular properties is stochastic, while biochemical relationships are deterministic."

Lines 484-485: The same as above. How do we choose the appropriate model and how do we know which model tells «the truth»?

Similar to the comments just above, it is our perspective that each null model provides a piece of the overall truth/reality of a given system under investigation. No null model or any other analysis can provide complete truth for any system. All analyses give us a digestible piece of information about a certain aspect of a system. Changing the null model will give different information because each null model is designed to 'extract' a certain kind of information. This is different than in microbial work in which some null models are not appropriate because the underlying assumption of phylogenetic signal is not upheld. These points have been clarified in the 4th paragraph of sub-section "Spatiotemporal variation in molecular properties is stochastic, while biochemical relationships are deterministic."

Lines 507-509: Yes but this makes sense if the concept of metacommunity makes sense in the examined ecosystem parts, i.e., if the spatially distinct samples within it are connected by dispersal. Is this true in the example dataset? I can understand it for the river water but I wonder if this is also true for the pore water.

There is significant hydrologic mixing of groundwater and surface water within the pore fluids of our field site, and there are very common directional reversals of surface water flowing into the pore fluids and pore fluids flowing into the river. As such, we conceptualize metabolites in the pore fluid to be part of a meta-metabolome that is connected through dispersal mediated by hydrologic transport and mixing. These hydrological connections are now called out (briefly) in the 1st paragraph of sub-section "An example set of metabolite assemblages and microbial communities."

Lines 524-525: Obviously not significant to the point of causing homogenization?

Mixing wouldn't lead to homogeneous selection, but rather homogenizing dispersal (a stochastic process). For the purposes of brevity, we didn't investigate the stochastic processes in-depth using the Raup-Crick until the discussion section.

Lines 530-533: Why would mixing affect only the MCD but not the TD? In the end what matters is the metabolome matrix that will tend to be more similar on average due to mixing irrespectively of the nature of the compounds (similarly to the effect of mass transfer events in metacommunity ecology).

Mixing could be detected when using either of these dendrograms, however, we only observed a significant impact of potential dispersal when using either the MCD or TWCD. We suppose that mixing might be less likely detected when using the TD, however, due to the relationships it measures. Whereas the gain or loss of a molecular formula might affect measurements made using the MCD/TWCD more significant, the redundancy in transformation paths between metabolites makes the gain or loss of a single metabolite less impactful. We completely expect that dispersal-based processes will be observed in other ecosystems or datasets when using the TD and are excited to see when and where these processes dominate. We see such patterns in our putative activity analyses as a potential case where stochastic assembly is observed when using the TD.

Your statement regarding the matrix is correct in that mass effects should allow any compound to transfer in space, however our view is slightly more nuanced. For example, some organic compounds are more likely to adsorb (as a function of their molecular formula) to minerals than others. Would this be considered stochastic (i.e., dispersal limitation) or should be it considered deterministic (i.e., an inherent trait promoting it to remain in one location)? This is similar to the discussion in microbial ecology about whether microbial movement can be strictly considered stochastic due to propensity of movement caused by flagella, propensity to form biofilms, etc. (Zhou and Ning, 2017 – MMBR).

Reviewer #3 (Remarks to the Author):

High-resolution mass spectrometric techniques are revolutionising our ability to understand biogeochemical cycles. The major advance of this paper is a framework to investigate the mechanisms underlying the spatiotemporal dynamics of metabolites by integrating concepts and tools developed in the study of biological communities. The authors of this study argue that ecological communities offer a strong conceptual analogue to metabolite assemblages, so the principles for studying them are directly transferable. I cannot agree enough! These points have been alluded to in previous studies (e.g. Mentges et al. 2017 *Front. Mar. Sci.*; Tanentzap et al. 2019 *PNAS*), but a unified synthesis is sorely needed. Therefore, the framework developed in this paper offers a major conceptual anchor and will be transformative for the immediate field and more widely in ecology and biogeosciences.

The authors demonstrate proof-of-concept for their framework in the context of predicting the processes governing the assembly of metabolite communities. They present two main analytical approaches to analysing assembly processes using molecular properties and biochemical transformations with either dendrogram or dendrogram-free null modelling. The advantages of these approaches, and the dendrogram specifically, are the ability to determine whether the communities (and their composition) arise simply because of chance or are different (e.g. more similar) than random, such as if they are structured deterministically. The specific results of these

analyses, applied to a small dataset from the Columbia River corridor, are not to be taken literally and I am far less concerned about what inferences we can draw in this specific example or even if they are "correct". For me, this paper is much more about the concepts. That said, I would have liked a bit more specific discussion on why resolving metabolite assembly dynamics is important.

Thank you for the encouragement. We have expanded discussion regarding the why understanding the assembly of molecular formula assemblages are important throughout the paper. For example, understanding metabolome assembly provides new opportunities to 'zoom in' to deterministic elements of metabolomes while potentially setting stochastic elements aside. This is useful for more robustly and mechanistically linking meta-metabolomes to microbes and, in turn, to biogeochemical function. Not all features of meta-metabolomes can or should be used in mechanistic models, and understanding assembly dynamics provides a way to focus attention on which meta-metabolomes and which part of meta-metabolomes are most likely to be important to consider in mechanistic models. This perspective has been added to the Abstract and the first paragraph of the Conclusion section.

I thought the work was technically sound, but I struggled to follow all the details, especially the estimation of the transformation-based dendrogram (TD), and I worry about its reproducibility. The TD is the minimum number of steps a given metabolite must take to reach another metabolite in the biochemical network, calculated for all pairwise comparisons. Yet it is not apparent to me how available these data are beyond this specific study system because the approach requires a database of putative biochemical transformations. Presumably the database used in this study (Supplemental File 4) is specific to the authors' observations and not entirely transferrable in other contexts. So if someone wanted to calculate TD afresh, how would they go about doing it? That information seems to be glossed over in the text and Supplementary Information. Somewhat related, I thought NRI and NTI say very different things about where differences lie in the dendrograms (e.g. roots vs tips; Webb et al. 2002 ARES), but the distinction is not explicitly presented in the text nor why these differences might matter. There is some attempt to address this point on lines 397 to 399, but I'm not sure it is clear enough and I think the statement on line 398 would be equally true of clustering towards the root.

The putative transformations were not derived from/for our study, but rather were compiled by one of our co-authors based upon known/common biochemical transformations. If someone wanted to perform the same analysis, they could use the same transformation database and infer mass differences based upon our scripts (or their own). We have added some text in the Supplemental Material to elaborate on the methodology. Specifically, the transformation database is likely to be expanded as we learn more about biochemical interactions. Fortunately, our methodology was intentionally designed to be compatible with future developments.

Regarding NRI and NTI, NTI is focused on tip-level clusters (i.e., closer toward the leaves) whereas NRI is focused on deeper-level clustering (i.e., closer toward root). Per your comment, we modified the text to capture insights derived from both NRI and NTI.

Related to this last point, I thought there were opportunities to improve the clarity and causal inference of the presented claims to strengthen the general conclusions and their impact. First,

like biological taxa, individual metabolites will undergo fluctuations in production and degradation. The authors claim this process is analogous to birth and death rates. But unlike biological taxa, these rates are much more intrinsically linked because the degradation process produces (i.e. gives birth) to new products. This is related to my comment on line 145 about "inheritance" of elemental composition. So I think the analogies between biological taxa and metabolite compounds can be strengthened in this vein. Second, the authors claim metabolites are governed by passive movement on line 128. But are they? Sure, metabolites themselves can't move. But their vectors can and do so in relation to other metabolite gradients. How would these interactions influence the framework?

We hadn't considered that perspective (namely, the vectors moving, not the metabolites) and have added some discussion in order to incorporate that. Metabolites are certainly governed by passive movement (diffusion and advection) but this highlights an ongoing discussion within ecology as to the degree which dispersal is stochastic in itself (Zhou and Ning, 2017 – MMBR). The overarching question is "Are the vectors moving stochastically, and how does such movement impact metabolite assembly?" – we believe that our methodology could be used to address this question. Associated text is in the 3rd paragraph of sub-section "Conceptual parallels between metabolites and ecological units." This is a really fascinating perspective and we would like to explore it further, though it is beyond what we can cover deeply in the present manuscript.

Regarding the inheritance component, we certainly agree with your overall sentiment. It is very interesting how metabolite 'death' is coupled inherently to metabolite 'birth.' As you note, that is distinct from a biological system, though if we think across trophic levels, death of an organism is always tied to the birth or at least sustenance of another. We incorporate a brief call out of the birth/death coupling in the second paragraph of sub-section "Conceptual parallels between metabolites and ecological units." This is an exciting conceptual direction that needs further development beyond what we can devote in our already-very-long manuscript.

Finally, the authors state that "opportunities to generate conceptual outcomes that would not be possible with traditional multivariate analyses". They then conclude that their analyses employing these methods demonstrate "that pore water metabolites (1) span a broader range of molecular properties and (2) are separated by a larger number of biochemical transformations." Won't "traditional multivariate analyses" reach these same conclusions? So I'm not sure this point about comparisons is what the authors really want to emphasise here. It seems like a false dichotomy.

In the specific case referenced in this comment, this only represents the conclusions from the alpha-diversity measurements. One of the key distinctions is that these are systematic differences – some environmental factor is either driving molecular properties to be over-dispersed or clustered. We have augmented the wording to emphasize this point, in the 6th paragraph of sub-section "Deterministic organization of metabolome α -diversity is revealed using among-metabolite relational information."

In addition, the broader point is well taken, and we have softened language throughout the manuscript to avoid the false dichotomy between traditional multivariate analyses and null

models. The key point is that null models are complementary to traditional methods in that they provide information on processes governing the assembly of meta-metabolomes.

A pedantic point that you should feel free to ignore is that I the meta-meta naming is pretty brain racking and unlikely to win over a general audience. I think it is etymologically sound, since a metabolome is an assemblage of metabolites from a sample and you have multiple assemblages in the environment, hence the "meta". And a similarity in definition can be drawn to genomics, where we have meta-genomics being the collection of genetic material from multiple environmental samples. But I think it gets somewhat muddled where metabolomes are defined on line 98 as the assemblages of metabolites from an ecosystem. So if metabolites are the assemblage, why do you need a meta? What does that differentiate? Assemblage of assemblages? I think if you are going to invent a word - which is fine - it just needs to be clearly defined. A further example of my confusion is on line 251 where we have "metabolome meta-assemblages". How is the meaning of this different from "metabolome assemblages". Both phrases mean groups of assemblages? The authors even conclude the paper by saying "metabolite assemblages are treated as analogs to ecological communities", so then why do you need the "meta"?

This is a very fair point – we’ve had long discussions regarding what term would best describe our methodology. The “meta” component is meant to invoke similarities to the meta-community concept, which is a set of local communities capable of interacting through dispersal (Leibold et al., 2004 – Eco Lett). In the case of “meta-assemblages” vs. “assemblages” we are trying to stress that we are developing a framework that enables the comparisons of multiple assemblages separated by some spatiotemporal scale, not only those that occur in some localized area. To help clarify we added a defining sentence at the end of the introductory paragraph. In addition, the phrase "metabolome meta-assemblages" has been changed to “meta-metabolomes” and the conclusion has been modified to refer to “meta-metabolomes.”

In sum, this is a terrific and inspirational foundation paper. I hope the authors see my comments as a reflection of my enthusiasm and desire to the broaden the uptake of their important conceptual work.

Andrew J. Tanentzap
ajt65@cam.ac.uk

Specific comments:

Line 69: Either "such as" or "like".

We have fixed this error (and structured the entire sentence to incorporate other MS techniques).

Line 101: Share should be plural.

“Share” has been changed to “shares”

Line 145: But they do pass on traits. They "evolve" as they are transformed through degradation processes and even pass on some of their properties, e.g. degradation products with P atoms will

only have them if the parent compounds had a P atom, no? I think this point is also relevant on line 643.

We agree with the overall sentiment which is why we posit the MCD as a viable way to track relationships. We feel, however, that traits are not inherently conserved as in a biological context. For example, within a group of obligate aerobic microbes, it is unlikely for a parent to give rise to an obligate anaerobe due to trait conservation. Within metabolites, however, some parent molecule that gives rise to a synthesis product with P atoms via phosphorylation doesn't necessarily have a P atom and therefore could participate in a realm of other transformations. Given the inconsistency of trait inheritance (there are cases where it holds true, cases where it doesn't), the analogy has to remain incomplete. We have added some text to the "Conceptual parallels" section to clarify our current thinking on this topic. It will be exciting to see how these ideas develop as more researchers follow and illuminate these kinds of conceptual paths.

Line 260: Replace "that" with "than"

We have change "that" to "than"

Line 264: It would be more useful here to have the biological explanation and the results phrased in terms of stochastic vs deterministic structuring.

We are not using NRI/NTI to really ascertain stochastic vs. deterministic structuring per se; that's more in the territory of bNTI. This isn't to say that NTI and NRI can't be used in this capacity, but we were just using these tools to understand the structure of the dendrogram.

Line 321: I struggled with this prediction as you can have stochastic and deterministic processes being important, but your analyses will only detect one of these for a single dataset and it is unclear which. I also think this prediction needs to be better justified. Why would a distinct environment lead to deterministic signatures per se? Also, you seem to be conflating assembly processes within communities here (e.g. deterministic signatures) with between-site differences (i.e. "we expect ... river water and pore water ... will lead to"). The concept of "variable selection" also wasn't clearly enough defined for me in biological terms at this point in the text.

bNTI is a pairwise metric calculated by comparing each sample to all other samples – the bNTI value will tell us whether the difference between any two samples was governed by stochastic or deterministic processes. As such, the method detects stochastic/deterministic processes per pairwise comparison. These comparisons (i.e., bNTI values) can be grouped and parsed based on science questions of interest. One can examine the whole dataset or examine comparisons just within, for example, pore water or exclusively between pore water and surface water.

Regarding the distinct environment comment, this prediction was motivated by previously studied differences in NOM within hyporheic zone environments and surface water in our field system (Graham et al., 2018 – STOTEN, Stegen et al., 2016, 2018 – Nat Comm). More broadly, the hyporheic zone is known to have divergent microbial communities and differential respiratory/metabolic capabilities (Danczak et al., 2016 – JGR Biogeo., Lynch et al., 2019 – Nat

Comm, Saup et al, 2019 – JGR Biogeo). We have added some text and citations to this sentence to elaborate our initial hypothesis.

Regarding the assembly processes vs. between-site differences comment – we identify which assembly processes give rise to observed variation by comparing sites. In this context, the notions are intertwined.

Regarding the variable selection comment, we have added some context to the paragraph first describing bNTI and the accompanying interpretations under the “Using metabolite dendrograms” section.

Line 334: I think you need to reconsider whether it is appropriate to reference the results here when you haven't presented them yet. I'm actually not overly concerned about the sample size - because like the authors say, the data are used for "demonstration purposes" and this paragraph could be moved later in the paper where the implications of the demonstration are discussed.

Fair point – we have removed the discussion of data but kept the note to ensure readers know that this data is more for demonstration purposes, than for actual conclusions.

Line 341: I think it would be appropriate to recognise here that there are studies that employ relational information to determine whether differences exist between samples like Mentges et al. 2017 Front. Mar. Sci; Tanentzap et al. 2019 PNAS.

Thank you very much for these citations - we have incorporated these perspectives here and throughout the manuscript.

Line 372: And what advances in biogeochemistry would answering these questions deliver? I think you need to add some text to make this benefit explicit.

As we mentioned in another comment, not all features of meta-metabolomes should be used in mechanistic models and understanding assembly dynamics provides a way to focus attention on important components. We have added some text in this paragraph to be more explicit.

Line 449: The description of the statistical analysis is incorrect here. NMDS is an ordination technique for dimensionality reduction and thus visualising differences in multivariate data. It cannot produce "Pseudo-F" values. I suspect the authors used something like a permutational analysis of variance here (PERMANOVA), but these details, along with the performance of the NMDS, do not appear in the main text or supplementary information.

Good catch– that is precisely what we did. We have added information regarding PERMANOVA to the supplemental methods.

Line 517: But were these values significantly different from that expected by random, e.g. +2? Isn't that what matters here?

There are two ideas at play in this statement. If $|bNTI| > 2$, then a deterministic process is indeed at play. However, per Stegen et al, 2015 – Front Micro, we know that bNTI contains metric-like qualities; when bNTI becomes increasingly large (positive), it indicates a strengthening of variable selection. That is, it is not just a binary ‘significant or not.’ In this case, there is information to be obtained by knowing that the pore water always has significantly higher bNTI than the surface water. This section has been edited to clarify these points.

Line 528: Should you cite Figure 5b here?

Yes – a reference to Figure 5b has been added.

Line 563: Please provide some evidence that 40 is a large number in your dataset, e.g. a histogram of number of transformations per molecule would show this point.

A supplemental figure has been provided to elaborate why we selected 40. We have also added text to explain that this was a fairly arbitrary number selected for demonstration purposes, rather than to conclude anything firm.

Line 566: Should processes be plural here to be grammatically consistent with vary?

Yes it should be – “process” has been replaced with “processes”

Line 599: Why would you expect this biologically? The idea needs fleshing out.

We added some text to flesh out the idea. If the microbial community shifts due deterministic processes, we expect that associated metabolite assemblages will also shift deterministically, but only if metabolite assemblages are governed by microbial composition and/or the same environmental factors influencing microbes. However, if microbes and metabolites are governed by different factors, deterministic assembly of one may not translate into deterministic assembly of the other.

Line 610: The conclusion for a "weak" association should come from the r value and not the p-values. P-values are either significant or not. There is no "weak" option as they shouldn't be interpreted as effect sizes. P-values are also biased by sample size in correlational tests. So you should ask if $r = 0.223$ is any less "weak" than $r = 0.298$... Who knows? And why dredge the data with both tests. Pick the most appropriate - likely the Spearman one given the small sample size.

The text has been revised based on only using Spearman, as suggested, which results in a non-significant relationship.

Line 613: It isn't "contrary" to your hypothesis per se. In a Popperian framework, failure to reject the null doesn't mean you accept it. Maybe you can't reject the null because you haven't measured the right things (like suggested on lines 621 onwards) or the sample sizes are small or the characterization of the assembly processes is incomplete.

We agree with this perspective and have changed the term to ‘inconsistent with.’ In our view, that is just a statement of the result (i.e., it didn’t go the way we hypothesized). We don’t use it to imply that we are accepting the null or that we believe the hypothesis will be rejected in other systems.

Line 624: Or relationships with functional diversity as we have done previously (see Tanentzap et al. 2019 PNAS). Of course, perhaps the most parsimonious explanation is that you haven't measured the right things. Does that point need to be countered?

We have added some extra discussion (including functional diversity) regarding what these results indicate as we believe there was some confusion regarding this section. We were not implying that a disconnect between the assembly processes reflects an inherent disconnect between the metabolome/microbial community, but rather that knowing when and where an assembly disconnect occurs is important. In this vein, we believe our revisions have addressed your comment as we believe the functional diversity approach is a complementary method to ours (it incorporates similar ideas, but uses a different metric). In addition, the text implies the need to measure other aspects of the microbial community (e.g., metatranscripts), which is consistent with the reviewer’s perspective that we haven’t measured the right things.

Line 638: But why does any of this matter? I think a general audience would appreciate some more specific exploration of why understanding coordination among assembly processes matters beyond being a fascinating aspect of biology.

This information will help provide insight into ecosystems and will highlight which components of the environment are the most significant. We have added a new sentence to the end of this section to expand on this concept.

Line 659: It help if the authors cited some of this "pioneering" work here.

A number of citations have been provided to highlight the history of using these methods to elucidate differences between these assembly processes.

Line 674: Why not just communities?

The goal is to draw parallels between metacommunity ecology and meta-metabolome ecology in that both are studying groups of communities/assemblages connected through spatial processes. We changed the language here to ‘meta-metabolome’ instead of ‘metabolite assemblages’ to emphasize this parallel.

Line 683: I think you are missing a word between "that" and "be". Furthermore, this point about disturbance kind of comes out of nowhere. Disturbance isn't mentioned anywhere in the paper yet somehow appears in the final take-home message? I would have thought the point is more general than disturbance, i.e. you can predict system responses full stop.

Fair point – we have removed “to disturbance” to keep the statement broad, and changed to ‘predict system dynamics.’

Line 899: You should define acronyms like AI_mod here.

Good catch – this information was once in the main text but got moved to the supplement. Acronym definitions have been added.

Line 907: Delete and just say something like "There were no significant differences between surface water and pore water samples as determined by a Mann Whitney U test." I also think this caption must explain what the boxes, vertical/horizontal lines, and points represent.

We have attempted to clarify the figure legend.

Line 922: Please specify the significance of the red dotted lines so the caption is self-contained. It might also help to specify that the p-values are for comparisons between pore-waters.

We have amended text in order to alleviate confusion.

Figure 1: I think for clarity it would help if you could connect the MCD and TWCD approaches by numbering the metabolite structures on the far left in a way that would be consistent with the network presented below. Also, it would help if you specified if the numbered metabolites are the only ones measured, whereas the empty circles are assumed but unmeasured intermediates. Finally, I think the TWCD path is a bit black box. Any way to summarise graphically how you combine the TD and MCD data?

We have modified Figure 1 per your recommendations and have amended the figure legend. The empty circles were not unmeasured intermediates but rather other identified peaks that we weren't addressing in the figure for brevity, but are addressed/accounted for in our actual analyses. This is noted in the revised figure caption.

Figure 3: I find it surprising there is no significant difference and thus p-value reported for the top panel in b) which looks very similar to bottom panel in b) and bottom in panel in a), which oddly seem to have identical p-values (presumably because of the permutational nature of the analyses with the small sample size)?

It's likely a function of relatively small sample size. The significant p-values are not far below the 0.05 threshold such that any small changes in the distributions can push the comparison to the non-significant side.

Figure 4: The only place the PCoA acronym appears in the whole paper is in this figure and its associated caption. Not only does it need to be defined but why are you mixing PCoA with NMDS? The text needs to give some gentle explanation here as it is not immediately obvious. The SI doesn't even mention why this is done.

We were just using these as visualization methods – in order to alleviate any confusion, we have removed the NMDS and replaced it with a Jaccard-based PCoA (and defined the acronym as well).

Reviewers' Comments:

Reviewer #1:

Remarks to the Author:

The revisions and responses by Danczak and colleagues were both thoughtful and appreciated. As the authors artfully proposed, the current manuscript was intended to act as a framework for future studies as the linkage between environmental metabolomes and biogeochemical processes are investigated in more detail. As such, my previous critiques regarding other environments or microbial assemblages were likely out of the scope of the current manuscript. However, the authors responded with additional citations and adjusted text to highlight the possible integration of other 'omics data and other systems in future studies. Clarifications to the statistics, of the processing of MS data, and additional figures added were also appreciated. In general, my questions were answered and the text is significantly enhanced and I look forward to seeing the impact of the current work in the future.

Remington X. Poulin, PhD

Reviewer #2:

Remarks to the Author:

In the revised version of the manuscript the authors have addressed all of my concerns. Moreover, I appreciate their remarks on the manuscript regarding the "infant" nature of the framework that is presented here for the first time, as well as the caveats that it comes with. This was exactly my main concern in the first manuscript version, but I think that now the readers are fully made aware of when and how this framework is applicable, what conclusions can, and cannot, be drawn from it, and how it can be further augmented. I only have a couple of remaining issues:

1) Lines 141-142 and 769: Homogenizing dispersal does not only occur due to mass effects; enough dispersal to balance differential selection can lead to homogenization. Please update the respective literature accordingly.

2) Lines 756-758: I do not understand how the current framework can highlight "parts of the meta-metabolome of importance". Unless the analyses are performed on an a-priori selected part of the metabolome as the authors demonstrated, the framework only provides sample-wise information. Am I missing something? Please rephrase to improve clarity.

Reviewer #3:

Remarks to the Author:

I thank the authors for thoughtfully addressing my previous comments. I found the manuscript much improved.

I just have a few remaining comments:

Line 167-168: I appreciate the response to my comment on this section in my original review but think this clause could still be fleshed out.

Line 284: I found these descriptions generally helpful but I'm not sure I fully understand what a "combined view" is. The previous text suggests this is "a point of view which is agnostic to a molecular formula's source" and "encompasses a putative biochemical point of view". Do those two necessarily reconcile?

Line 296-300: I'm kind of unclear about what is going on here. Why "must" a test for phylogenetic signal be performed? What if you have opposing processes in different parts of your tree, e.g. trait conservatism vs convergence. Won't that cancel out any signal? So then you won't have it "confirmed"? I thought the signal *is* "estimating phylogenetic structure" by definition. In fact, this is

what line 301 goes on to say - that signal estimates structure. So the wording on lines 299-300 that only when signal is confirmed can you estimate phylogenetic structure is confusing.

Line 303: Not sure I follow how the "niche values" are easier to estimate than traits. A bit more explanation here would help, perhaps by specifying what you mean by "niche values"?

Line 376: Remove the extra space after the comma.

Lines 704-706: Can you add just a few more words on what is being compared with the Mantel test? Wasn't entirely obvious to me. And, as I discussed in my previous review, the significance is ultimately down to sample size. Presumably the sample size is quite small to have a non-significant p-value with a Mantel $r = 0.22$? Can the n value be added to the statistics in parentheses?

Line 747: Perhaps "evolve along the same lines as biological species" to indicate as discussed previously there is some "trait" inheritance, it just differs from biological species in some ways.

Figure 5: Are you sure this plot is correct? The boxplots for TD in panel b) effectively show that all values aside from one outlier are >2 but a substantial amount of the TD density in panel a (maybe 30%?) looks to be beneath the dashed red line denoting 2.

Response to Reviewers

Reviewer #1 (Remarks to the Author):

The revisions and responses by Danczak and colleagues were both thoughtful and appreciated. As the authors artfully proposed, the current manuscript was intended to act as a framework for future studies as the linkage between environmental metabolomes and biogeochemical processes are investigated in more detail. As such, my previous critiques regarding other environments or microbial assemblages were likely out of the scope of the current manuscript. However, the authors responded with additional citations and adjusted text to highlight the possible integration of other 'omics data and other systems in future studies. Clarifications to the statistics, of the processing of MS data, and additional figures added were also appreciated. In general, my questions were answered and the text is significantly enhanced and I look forward to seeing the impact of the current work in the future.

Remington X. Poulin, PhD

Reviewer #2 (Remarks to the Author):

In the revised version of the manuscript the authors have addressed all of my concerns. Moreover, I appreciate their remarks on the manuscript regarding the "infant" nature of the framework that is presented here for the first time, as well as the caveats that it comes with. This was exactly my main concern in the first manuscript version, but I think that now the readers are fully made aware of when and how this framework is applicable, what conclusions can, and cannot, be drawn from it, and how it can be further augmented. I only have a couple of remaining issues:

1) Lines 141-142 and 769: Homogenizing dispersal does not only occur due to mass effects; enough dispersal to balance differential selection can lead to homogenization. Please update the respective literature accordingly.

In this specific case, that would still be homogenizing dispersal given that dispersal is overcoming the effects of selection. By the definition of dispersal that we have provided here, dispersal is a mass effect.

2) Lines 756-758: I do not understand how the current framework can highlight "parts of the meta-metabolome of importance". Unless the analyses are performed on an a-priori selected part of the metabolome as the authors demonstrated, the framework only provides sample-wise information. Am I missing something? Please rephrase to improve clarity.

This is correct – by selecting groups of metabolites *a priori*, you can identify components that might be more or less important to the metabolome structure.

Reviewer #3 (Remarks to the Author):

I thank the authors for thoughtfully addressing my previous comments. I found the manuscript much improved.

I just have a few remaining comments:

Line 167-168: I appreciate the response to my comment on this section in my original review but think this clause could still be fleshed out.

We have added two examples where

Line 284: I found these descriptions generally helpful but I'm not sure I fully understand what a "combined view" is. The previous text suggests this is "a point of view which is agnostic to a molecular formula's source" and "encompasses a putative biochemical point of view". Do those two necessarily reconcile?

Given that the putatively biochemical point of view is also source agnostic (e.g., we aren't claiming the compounds come from or go to any specific end point), we believe the combination is acceptable. We have reworded this section to emphasize an integration rather than a combination.

Line 296-300: I'm kind of unclear about what is going on here. Why "must" a test for phylogenetic signal be performed? What if you have opposing processes in different parts of your tree, e.g. trait conservatism vs convergence. Won't that cancel out any signal? So then you won't have it "confirmed"? I thought the signal *is* "estimating phylogenetic structure" by definition. In fact, this is what line 301 goes on to say - that signal estimates structure. So the wording on lines 299-300 that only when signal is confirmed can you estimate phylogenetic structure is confusing.

A phylogenetic signal occurs when phylogenetic structure approximates some trait distribution. Phylogenetic null modeling is only useful in identifying selective processes when traits are captured by the phylogeny/dendrogram. Given that a phylogenetic signal is analyzed at all distances within a tree, we are able to identify at which point in a tree trait conservatism occurs – where the phylogenetic signal occurs is important in informing which null model to use (i.e., conservation over short distances is required for bNTI). Given that a phylogenetic signal is not required for a trait-based dendrogram as we are using in this study, however, we do not need to identify one. Additionally, "estimating phylogenetic structure" was added in error and has been removed.

Line 303: Not sure I follow how the "niche values" are easier to estimate than traits. A bit more explanation here would help, perhaps by specifying what you mean by "niche values"?

We have expanded this section slightly to highlight the difference between niche values and trait values, but do not think it necessary to expand this discussion further given that a phylogenetic signal nor niche value are necessary for our analyses.

Line 376: Remove the extra space after the comma.

We have removed the comma.

Lines 704-706: Can you add just a few more words on what is being compared with the Mantel test? Wasn't entirely obvious to me. And, as I discussed in my previous review, the significance is ultimately down to sample size. Presumably the sample size is quite small to have a non-significant p-value with a Mantel $r = 0.22$? Can the n value be added to the statistics in parentheses?

We were comparing the pairwise bNTI values generated by during both metabolomic and microbial null modeling. We have expanded this section to highlight specifically which values we were correlating. In the case of this test, the sample sizes were identical in both the metabolomic and microbial datasets because they were matched before the correlations could be run. Importantly, a Spearman-based

Mantel r -value = 0.22 is actually quite low, so the correlation would be weak even if it was significantly correlated.

Line 747: Perhaps "evolve along the same lines as biological species" to indicate as discussed previously there is some "trait" inheritance, it just differs from biological species in some ways.

We have incorporated your suggestion so that it now reads "(i.e., metabolites do not evolved like biological species).

Figure 5: Are you sure this plot is correct? The boxplots for TD in panel b) effectively show that all values aside from one outlier are >2 but a substantial amount of the TD density in panel a (maybe 30%?) looks to be beneath the dashed red line denoting 2.

The plot is accurate. Panel a is plotting all pairwise comparisons for a given dendrogram while panel b) is only plotting the average bNTI value for within-group comparisons (i.e., within-pore water or within-river water only); these differences lead to the distinct bNTI values. We have modified the figure legend to highlight this distinction more clearly.